# Trends in observed surface solar radiation and their causes in Brazil in the first two decades of the 21st century.

Lucas Ferreira Correa[1], Doris Folini[1], Boriana Chtirkova[1], and Martin Wild[1]

[1]Institute for Atmospheric and Climate Sciences, ETH Zurich, Zurich, Switzerland.

*Correspondence to*: Lucas Ferreira Correa (lucas.ferreira@env.ethz.ch)

**Abstract.** Numerous studies have investigated the long term variability of surface solar radiation (SSR) around the world. However, the large disparity in the availability of observational data between developed and least developed/developing countries leads to an underrepresentation of studies on SSR changes in the latter. This is especially true for South America, where few observational studies have investigated the SSR trends, and usually only at a local or regional scale. In this study we use data from 34 stations distributed throughout all the regions of Brazil to present the SSR trends in the first two decades of the 21st century and investigate their associated causes. The stations were grouped into 8 composites according to their proximity. Our results show that in the North and Northeast Brazil a strong dimming occurred, with significant contributions from increasing atmospheric absorption, most likely due to anthropogenic emissions, and increasing cloud cover. In the Southeast and Midwest regions of Brazil near-zero trends resulted from competing effects of clear-sky processes (attenuation of solar radiation under cloudless conditions) and strong negative trends in cloud cover. In the South part of the Amazon and in Southern Brazil a statistically insignificant brightening was observed, with significant contribution from decreasing biomass burning emissions in the former and competing minor contributions in the latter. These results can contribute to deepen the knowledge and understanding of SSR long-term trends and their causes in South America, reducing the underrepresentation of this continent when compared to regions like Europe.

## 1. Introduction

Decadal trends in surface solar radiation (SSR) have been the subject of study since pioneering studies in late 1980s and early 1990s made efforts to try to understand the long-term variation of SSR (Ohmura and Lang, 1989; Russak, 1990; Dutton et al., 1991; Stanhill and Moreshet, 1992). Several studies have followed presenting the trends and discussing their causes and potential consequences in several parts of the world (Wild, 2009), such as in Europe (e.g. Natsis et al., 2024; Kazadzis et al., 2018; Manara et al., 2016; Norris and Wild, 2007; Power, 2003), North America (e.g. Liepert 2002), China (e.g. Feng and Wang, 2019; Wang et al., 2015; Xia et al., 2007), Japan (e.g. Kudo et al., 2012) and New Zealand (Liley, 2009). Global dimming (negative trends in SSR) and brightening (positive trends in SSR) have been associated, in most of the cases, with changes in cloud cover (e.g. Stjern et al., 2008;

Augustine and Capotondi, 2022) and changes in aerosol loadings (e.g. Wild et al., 2021, Kambezidis et
al., 2012), with the dominant aspect depending on regional atmospheric and emission features.
However, many regions of the world are still underrepresented by such studies, mostly because of the
lack of observational high quality data in most of developing and least developed countries in contrast
to regions like Europe, North America or Eastern Asia. South America is an important region to be
mentioned in this context.
The lack of long-term SSR data in South America, reported by different authors (Ohmura,
2009; Gilgen et al., 2009), is the main cause for the absence of a long literature in the region. Still, a
few studies tried to assess SSR variability in South America. Ohmura (2009) presented and discussed
SSR decadal trends based on a few stations in Venezuela at the end of the 20th century. Schwartz (2005)
used astronomical extinction measurements to estimate clear-sky SSR trends at one astronomical
observatory in Chile during two decades (1978-1997). Yuan et al. (2021) and Jiao et al. (2023) used
machine learning methods to spatially interpolate SSR ground observations and reanalysis data and
used this approach to assess SSR decadal variability over the whole globe, including South America,
covering the second half of the 20th century and the first two decades of the 21st century. Da Silva et
al. (2010), de Jong et al. (2019) and de Lima et al. (2019) all assessed SSR variability in Brazil with a
focus on the potential for photovoltaic energy production. Zuluaga et al. (2021) and Raichijik (2012)
used sunshine duration to assess the SSR variability in Brazil for the last 2 to 4 decades of the 20th
century and the beginning of the 21st century. A similarity between most of these studies is the fact that
they had to rely on reanalysis, modeling data and indirect estimators of SSR (like sunshine duration),
with the only of the abovementioned studies that used ground observations being limited to a small
region in Venezuela. This leaves regions such as the densely populated southeastern Brazil or the highly
climate-relevant Amazon region without any direct assessment of the regional SSR long term
variability. Yamasoe et al. (2021) presented and discussed a SSR time series of fifty-six years (1961-
2016) measured in the city of Sao Paulo, and that is, to our knowledge, the longest and most detailed
analysis of directly observed SSR in South America. The studies referenced here apply different
methods, to different regions, in different periods, so it is hard to directly compare them. But, in general
terms, studies based on sunshine duration tend to indicate a brightening in Brazil after 1980s, while
studies using machine learning techniques and regional observational studies show a spatial
heterogeneity in the SSR trends in Brazil in the last few decades. All these studies provide different
pieces of information about SSR variability in this part of the world, however, none of them provide a
large scale assessment of the long term SSR decadal trends using ground observations of SSR, as done
for regions like Europe (e.g. Chiacchio and Wild, 2010; Pfeifroth et al., 2018), China (e.g. Yang et al.,
2018) or the United States (e.g. Long et al., 2009).
To try to tackle this gap in literature, we made use of the availability of SSR data from
automated meteorological stations from the Instituto Brasileiro de Meteorologia (INMET) from 2001
onwards to provide a large scale assessment of SSR decadal trends and underlying causes at the
beginning of the 21st century in the Brazilian territory, which covers approximately half of the South
American continent. The direct assessment of SSR long-term variability (using observed SSR) over
such a large area in South America represents a novel contribution from this work. The objective of this
study is to present the in-situ observed SSR decadal trends around Brazil in the first two decades of the
21st century and discuss their underlying causes. This is done at the regional level, rather than locally,
by selecting stations in strategic locations around the Brazilian territory and grouping them into station
composites. With this study we intend to help to reduce the under representativity of Global Dimming
and Brightening (GDB) studies in South America.
**2. Data and methods**
**2.1 In situ SSR and cloud cover measurements**
Surface solar radiation data for 32 of the 34 stations (see table 3, in annex) is collected and
controlled by the Instituto Nacional de Meteorologia (INMET) and was retrieved from the BDMEP
portal (available at: https://bdmep.inmet.gov.br/ (last access 27 Oct 2023)). The stations were chosen
based on data availability in the regions of each composite used in this study (see section 2.4). The data
was retrieved at hourly time resolution. All data was tested at the hourly time scale for consistency
using the physical and extremely rare limits established by Long and Dutton (2002). None of the
INMET stations used in this study were reported to have major discontinuities in the records.
Nevertheless, we still applied the penalized maximal F test by Wang (2008) to verify the time series for
inhomogeneities. No homogeneity problems were identified in the stations used in this study.
The hourly values were further converted into daily means by simply taking the average of the
24 hourly values in a day. If one hourly value was missing (due to either lack of data or removal during
quality test) the one hourly value was filled linearly using the previous and next hours and the daily
value was the average of 24 hourly values (23 observed and 1 filled linearly). If more than one hourly
value was missing, the daily value was not calculated. In this way, daily values were always the result
of the average of 24 hourly values. Daily values were further converted into monthly values by simply
averaging the daily means within the same month. Monthly values were only calculated when at least
70% of the days in a month were available. For example, 70% of the days of any April are equivalent
to 21 days. Therefore, if any April had 21 or more valid daily values, the monthly value would be
calculated as the average of all available daily values. On the other hand, if less than 21 days had valid
data, the monthly value would not be calculated. Further conversion from monthly to annual values
again occurred by simply averaging the 12 months. If one, two or three monthly values went missing,
the long term mean (mean for the whole period with available data) for that month would be used
instead, and the annual mean was still calculated. If more than three monthly values were missing, then
the annual value was not calculated. In this way, the annual value was always a result of 12 monthly
values, from which no more than 3 were filled with the long term mean. The averaging procedure from
daily to monthly, and from monthly to annual values reproduces similar methodologies used in previous
studies (e.g. Stjern et al., 2008; Manara et al., 2016).
108      The BSRN (Baseline Surface Radiation Network, Ohmura et al., 1998; Driemel et al., 2018)
station at Florianopolis was also used in this study. Its data was provided at 15-minute intervals. Data
from the station operated by the Instituto de Astronomia, Geofisica e Ciencias Atmosfericas of the
Universidade de São Paulo (IAG/USP), located in the city of Sao Paulo was also used. Data from this
station was provided as daily means. Both time series were also checked for consistency with the same
procedure applied to the INMET stations, at the hourly time scale for the BSRN station and at the daily
time scale for the IAG/USP station. Metadata for both stations did not report any discontinuities, and
the tests performed using the penalized maximal F test by Wang (2008) also did not indicate any
inhomogeneity in the time series. The SSR long term variability at the Sao Paulo station was previously
carefully analyzed by Yamasoe et al. (2021). This station also has the longest time coverage among all
of the stations used in this study: all the other stations only have data after 2000, while this stations has
available data from decades earlier. But we limited the analysis to the period with coverage of the other
stations because we intend to investigate the SSR variability at the regional level (composites) rather
than at the local level (individual station). The procedure to convert from sub-daily to daily averages,
from daily to monthly and from monthly to  annual values at these two stations was the same as the
procedure used for the INMET stations.
124      Cloud cover data was also retrieved through the BDMEP portal from the INMET. The stations
were the same as used for the INMET SSR measurements with the addition of data from Florianopolis.
In Florianopolis, where the SSR data is originally from BSRN, the location of the SSR and the cloud
measurements differ by a few kilometers. Cloud cover data is collected from visual inspections at 00,
12 and 18 UTC and is provided in units of tenths (1/10) of cloud cover.. The daily cloud cover values
used in this study are a result of the average from the 12 and 18 UTC observations. This is equivalent
to 9 and 15 local time at most of the stations used in this study (8 and 14 for the westernmost stations).
At the Sao Paulo station, the diurnal cloud cover values are a result of hourly observations between 7
and 18 local time. Cloud cover data was converted into monthly and then annual values using the same
procedure as used for the SSR data. The cloud cover data is also used to calculate the Cloud Cover
Radiative Effect (CCRE), following the procedure described by Norris and Wild (2007). This variable
gives an estimation of the change in SSR produced by changes in cloud cover.
136      The SSR data described in this section is used to estimate the SSR trends presented in table 1,
and to calculate the fractional atmospheric column absorption (see section 2.4), which also has the
trends presented in table 1. The cloud cover data described in this section was used to estimate cloud
cover trends presented in table 1 and to apply one of the two methods for clear-sky identification used
in this study (see section 2.3).

141      **2.2 Satellite and reanalysis data**

To investigate Aerosol Optical Depth (AOD) variability, we used data from the CAMS
(Copernicus Atmosphere Monitoring System) reanalysis (Inness et al., 2019), provided by ECMWF.
This product has monthly time steps and spatial resolution of approximately 80 km, with temporal
coverage starting from 2003. Gueymard and Yang (2020) validated CAMS data using AERONET
stations from around the world, including South America and found that the reanalysis performs well
in comparison to in-situ aerosol observations, therefore being well suited for regional and global studies.
To assess the Aerosol Absorption Optical Depth (AAOD) at 500 nm we used data from the OMAERUV
aerosol algorithm from the Ozone Monitoring Instrument (OMI, Torres et al., 2007). The product is
provided at daily time resolution and 1-degree resolution, and is available from 2004 onwards. Due to
the frequent occurrence of missing daily values in the AAOD data from OMI (due to different aspects,
such as cloudy scenes), conversion from daily to monthly values was done only when at least two days
in a month were available. From monthly to annual values the conversion was only performed when at
least 11 of the 12 months had available data (missing month would be filled with long term mean, that
is, the mean for the whole period with data availability). We should also highlight that aerosol
absorption is a variable highly dependent on the spectral region, thus the absorption at 500 nm could
not be representative for the whole spectrum.
We also used shortwave radiative fluxes measured at the Top of the Atmosphere (TOA) by the
CERES (Cloud and Earth's Radiant Energy System, Doelling et al., 2013) instruments on board of the
satellites Terra and Aqua. The CERES-SSF product (Doelling et al., 2016), used in this study, provided
TOA shortwave fluxes at monthly time intervals and 1-degree spatial resolution, from 2000 onwards.
The same product also provided incoming shortwave radiative fluxes at the TOA, which was also used
in this study. The data from CERES was used to estimate fractional atmospheric column absorption
(see section 2.4).
Anthropogenic emissions were assessed using EDGAR (Emissions Database for Global
Atmospheric Research, Crippa et al., 2018). The data provides anthropogenic emission estimates at 0.1
degree spatial resolution and does not consider large scale biomass burning, land use change and
forestry (Crippa et al., 2018). This dataset was used, even though it does not include biomass burning,
because it provides information about aerosol emissions from all other sources, which are also relevant,
such as urban and industrial emissions. For this study we acquired the data in annual values and in units
of $kg\ m^{-1}\ s^{-1}$. The unit was further converted to $kg\ grid^{-1}\ year^{-1}$ (kg emitted for each 0.1 degree grid per
year). Finally, total column water vapour was obtained from the ERA5 reanalysis (Hersbach et al.,
2020), which provides data with a 0.25 degree spatial resolution and monthly time resolution. Cloud
cover from ERA5 was also used as supporting information in addition to the previously mentioned
SYNOP cloud cover, measured in-situ.
The AOD, AAOD, water vapor and anthropogenic emissions data described in this section were
used to identify the spatial distribution of the trends for these variables. The TOA incoming and
outgoing irradiance data described in this section was used to estimate fractional atmospheric column
absorption (see section 2.4). For all gridded data described in this section, the stations were sampled by
taking the grid box containing the station coordinates.

**2.3 Clear-sky SSR**
Time series of clear-sky SSR were derived using two different methods. At all stations we used
(1) the clear-sky method proposed by Correa et al. (2022), and at the stations with Synop cloud cover
data we also (2) derived clear-sky using cloud cover information. We applied both methods on the daily
time series. For the first method, we calculate station specific daily transmittance thresholds for every
month of the year. Days with transmittance lower than this threshold for the specific station in the
respective month are flagged as cloudy and removed. Days with transmittance above the thresholds are
flagged as clear-sky. As this method relies on the reduction of atmospheric transmittance under cloudy
conditions, its main weakness is associated with extreme aerosol events that could suddenly strongly
reduce transmittance. Thus this method is not well suited for the analysis of high frequency (interannual)
variability, but it has been shown adequate for assessment of long term trends (Correa et al., 2022).
For the second method, we simply used in-situ observations of cloud cover to identify cloudy
scenes. We set the threshold of cloud cover to two tenths (20%), where any day with cloud cover above
that was flagged as cloudy and removed. The choice of the cloud cover threshold represents a trade-off,
where low thresholds (say, 0%) would completely avoid any cloud signal but would also remove days
with low cloud occurrence, where the effects of cloud-free processes still dominate, and leave the time
series with very few valid values. For this reason, we allowed a higher threshold, assuming that on days
with such low cloud cover (0-20%) the cloud-free processes still dominate the signal of the SSR
variability.
In both methods, the removal of cloudy days results in clear-sky SSR time series with many
gaps. Thus, special care should be taken when converting from daily to monthly values and from
monthly to annual values. Monthly values were only calculated when at least two daily values were
available for the respective month. But before taking their average, each available daily value is
normalized to the 15th day of the month by multiplying the daily irradiance with a normalization factor.
This normalization factor is a result of the ratio between the TOA daily irradiance at the 15th day of the
month and at the day flagged as clear-sky. This is done to correct for the solar geometry at different
times of the month. From monthly to annual values the procedure is the same as done for all-sky SSR:
the calculation is done when at least 10 months are available, with missing values being replaced by
long term means. When less than 10 months are available, the annual means are not calculated.

**2.4 Fractional atmospheric column absorption**
The daily fractional atmospheric column absorption ($F_{abs}$) was calculated for every station by
combining SSR measured at the surface, surface albedo from ERA5 at the 0.25x0.25 degree spatial
resolution and incoming and outgoing shortwave radiation at TOA from CERES-SSF 1deg from the
Terra satellite (1x1 degree spatial resolution) and daily time resolution. For the gridded data the pixel
containing the station coordinates was used. These variables were combined in equation 1 to calculate
$F_{abs}$.
$$F_{abs} = 1 - (SW_{upTOA}/SW_{downTOA}) - ((1\text{-albedo}_{SFC})*(SW_{downSFC}/SW_{downTOA})) \qquad (1)$$
$SW_{upTOA}$ is the outgoing shortwave radiation at TOA, $SW_{downTOA}$ is the incoming shortwave
radiation at TOA, $albedo_{SFC}$ is the surface albedo and $SW_{downSFC}$ is the SSR. Thus, the term
$(SW_{upTOA}/SW_{downTOA})$ represents the fraction (0-1) of the incoming shortwave radiation at the TOA
which is reflected back to space, and the term $((1\text{-albedo}_{SFC})*(SW_{downSFC}/SW_{downTOA}))$ represents the
fraction (0-1) of the incoming shortwave radiation at the TOA which is absorbed at the surface. Then,
$F_{abs}$ represent the fraction of the incoming shortwave radiation at TOA which is absorbed within the
atmosphere column. $F_{abs}$ values can range between 0 and 1, where 0 would represent no atmospheric
absorption and 1 would represent a black body absorption by the atmosphere.

**2.5 Selection of station composites**
The stations used in this study were divided into eight composites based on geographical
proximity, demographics and atmospheric features found in Brazil. That is, the composites were
organized with the intent of covering different climate characteristics around the country and, in most
cases, included data from big cities (> 1 million inhabitants). The use of data from big cities facilitates
the construction of the composite time series, since the stations with longest time series and less missing
data were found near big centers. The composites are: [1] Manaus region, [2] Belem region, [3]
Fortaleza region, [4] Salvador region, [5] South Amazon, [6] Midwest Brazil, [7] Southeast Brazil and
[8] South Brazil. The location of all stations are shown in Figure 1, and colors and markers denote the
different composites. Each composite is composed of three to five stations. Based on literature review,
Reboita et al. (2010) divided the precipitation regimes in South America in 8 regions, out of which five
regions are in the Brazilian territory. Ferreira and Reboita (2022) revisited the topic and applied a non-
hierarchical clustering technique to classify the precipitation regimes in South America. The authors
also found 8 different precipitation regimes in the continent and only minor spatial differences to the
previous study, with five of the regimes being present in the Brazilian territory. All of them were at
least partly represented by the composites.

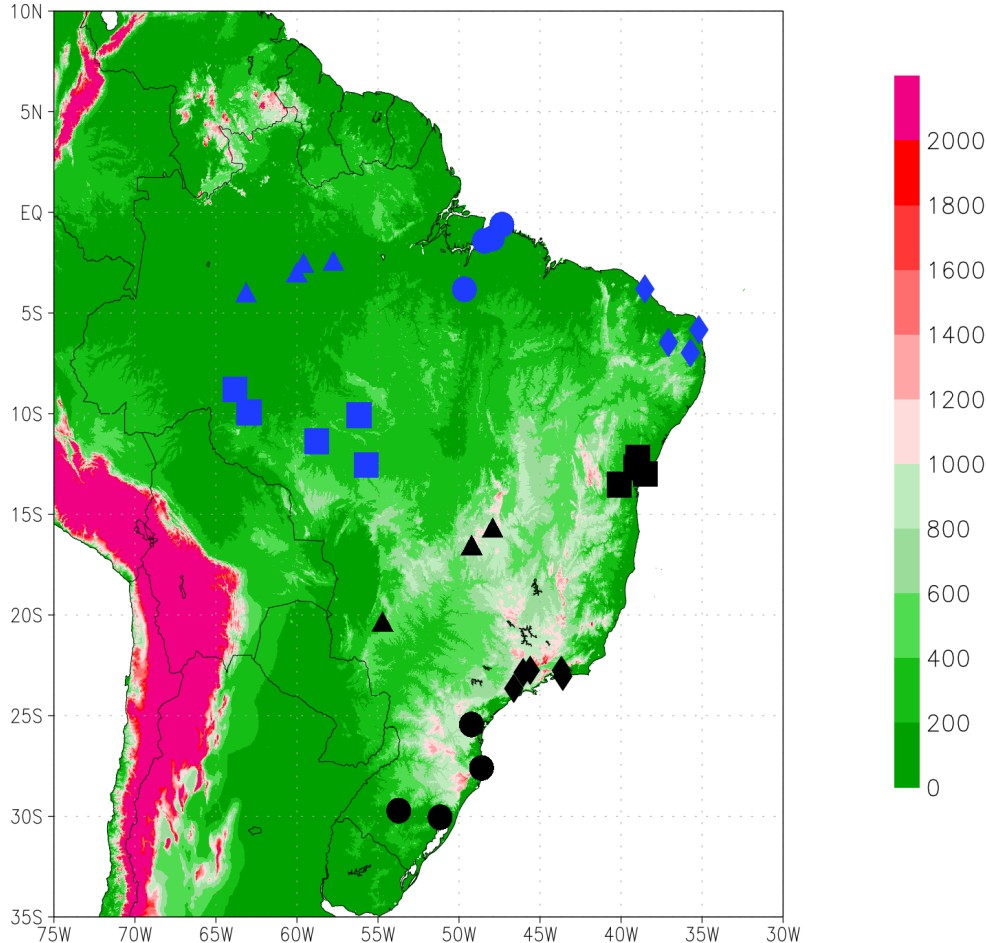

**Figure 1: Map of Surface Solar Radiation stations and composites used in this study and topography of**
**South America (in meters above sea level). Colors and shapes represent the different composites: Blue**
**triangles = Manaus region; Blue circles = Belem region; Blue diamonds = Fortaleza region; Black squares**
**= Salvador region; Blue squares = South Amazon; Black triangles = Midwest Brazil; Black diamonds =**
**Southeast Brazil; Black circles = South Brazil.**
In the north of Brazil two composites were centered around the two biggest cities in the
Brazilian Amazon, (1) Manaus and (2) Belem. Precipitation and cloudiness in both regions is strongly
tied to local to mesoscale phenomena, like local convection, sea breeze circulation and squall lines. At
the large scale, the Intertropical Convergence Zone (ITCZ) also has a significant influence on the
precipitation in the regions, playing a major role for the seasonality of precipitation (Fisch et al., 1998).
Feedbacks with the Amazon rainforest are also important, especially the recycling of precipitation. But
regarding biomass burning in the Amazon, the most important area is located in the southern part of the
Amazon (Artaxo et al., 2006), south of both Belem and Manaus. The occurrence of the South American
Low Level Jet (Vera et al., 2006), important for moisture and aerosol transport from the Amazon to
Southeastern Brazil, leaves the locations of Belem and Manaus with lower influence of biomass burning
aerosols than the southern fraction of the Amazon. Still, the influence of aerosols from the forests (either
biogenic or biomass burning related) should not be neglected (Rosario et al., 2019), and most
importantly, the importance of anthropogenic emissions from such big population centers should be
taken into account.
In the northeast of Brazil, the composites of (3) Fortaleza and (4) Salvador share similar general
characteristics regarding precipitation and cloudiness regimes. The stations in these composites are also
centered around big population centers (Fortaleza and Salvador), where anthropogenic emissions
should be taken into account. The biggest difference to the composites around Manaus and Belem, is
that these two composites are not located in the Amazon region. But they are located in the same
precipitation regime division proposed by by Ferreira and Reboita (2022), with two stations of the
Fortaleza composite being located in a different subdivision.
The composite (5), South Amazon, was chosen to cover the region under the strongest influence
of biomass burning aerosols from the Amazon (Artaxo et al., 2006). The stations in this composite are
located in a different subdivision by Ferreira and Reboita (2022), where large scale phenomena (such
as the Bolivian high, the South Atlantic Convergence Zone and cold fronts) play an important role for
the cloud formation. This composite is not centered around a big city, and the most populated city in
the area is Porto Velho, with a population of approximately 500'000 people (IBGE, 2022). A few
degrees south of the South Amazon composite, are the stations of the (6) Middle West Brazil composite.
They are located approximately halfway between the South Amazon composite and the densely
populated Southeast Brazil. It is a dry region mostly influenced by large scale phenomena, compared
to the north and northeast regions of Brazil. The biggest city in the composite is Brasilia.
The Southeast is the most densely populated area in Brazil, where big centers like Sao Paulo
and Rio de Janeiro are located. Like the Middle West and South Amazon composites, cloud formation
in this region is mostly associated with large scale phenomena, with significant influence from local
convection and sea breeze being limited mostly to summer months (Reboita et al., 2010; Ferreira and
Reboita, 2022). The (7) Southeast Brazil composite covers this area. The transport of humidity and
aerosols from the south Amazon are both relevant aspects to consider. But regarding aerosols, urban-
industrial emissions from the large population centers should be more relevant. The last station
composite covers the Southernmost part of the country. The (8) South Brazil composite is entirely
located in subtropical latitudes, and covers its own precipitation regime subdivision from Ferreira and
Reboita (2022). Large scale phenomena like frontal systems and extra-tropical cyclones play a major
role for cloud formation and moisture transport from the ocean. It is also a densely populated region,
with big cities like Porto Alegre, Curitiba and Florianopolis, thus, urban-industrial aerosol emissions
should be taken into account.
The whole discussion in this study revolves around these eight composites. Each variable was
fully processed and converted to annual values at the station level, and only after that, they were grouped
with the other stations in the respective composite. The list of stations in each composite can be found
in Table 3 (in appendix).
**2.6 Trend calculations**

The trend analysis was based on annual anomalies of SSR. To calculate the annual anomalies,

the absolute SSR annual values were subtracted from the average SSR value for the whole period of
data availability for the respective composite (see table 1). This did not affect the trends, but facilitated
the visualization and comparison between time series, since the anomalies are centered around a
common value (zero). Decadal trends of SSR and most other variables presented in this study were
calculated using a Linear Least Squares (LLS) regression, with the confidence intervals (at the 95%
confidence level) being calculated using equation 4 from Nishizawa and Yoden (2005). Cloud cover
time series, in most cases, did not have the residuals normally distributed, thus, to account for that, we
calculated their trends using the Sen's slope (Sen, 1968) and Mann-Kendall test (Mann, 1945; Kendall,
1975). Trends of ground observations were calculated for the whole time availability of the composites,
however, as the time availability varies from one composite to the other, the periods used for trend
calculations vary by a few years. SSR trends are displayed in units of $W/m^2$ per decade. The period
considered in each composite is displayed in Table 1.

**3. Results**
**3.1 All-sky and clear-sky SSR trends**

Figure 2 shows the all-sky SSR anomalies time series of the 8 composites analyzed in this

study. All trends calculated in this study are shown in Table 1.

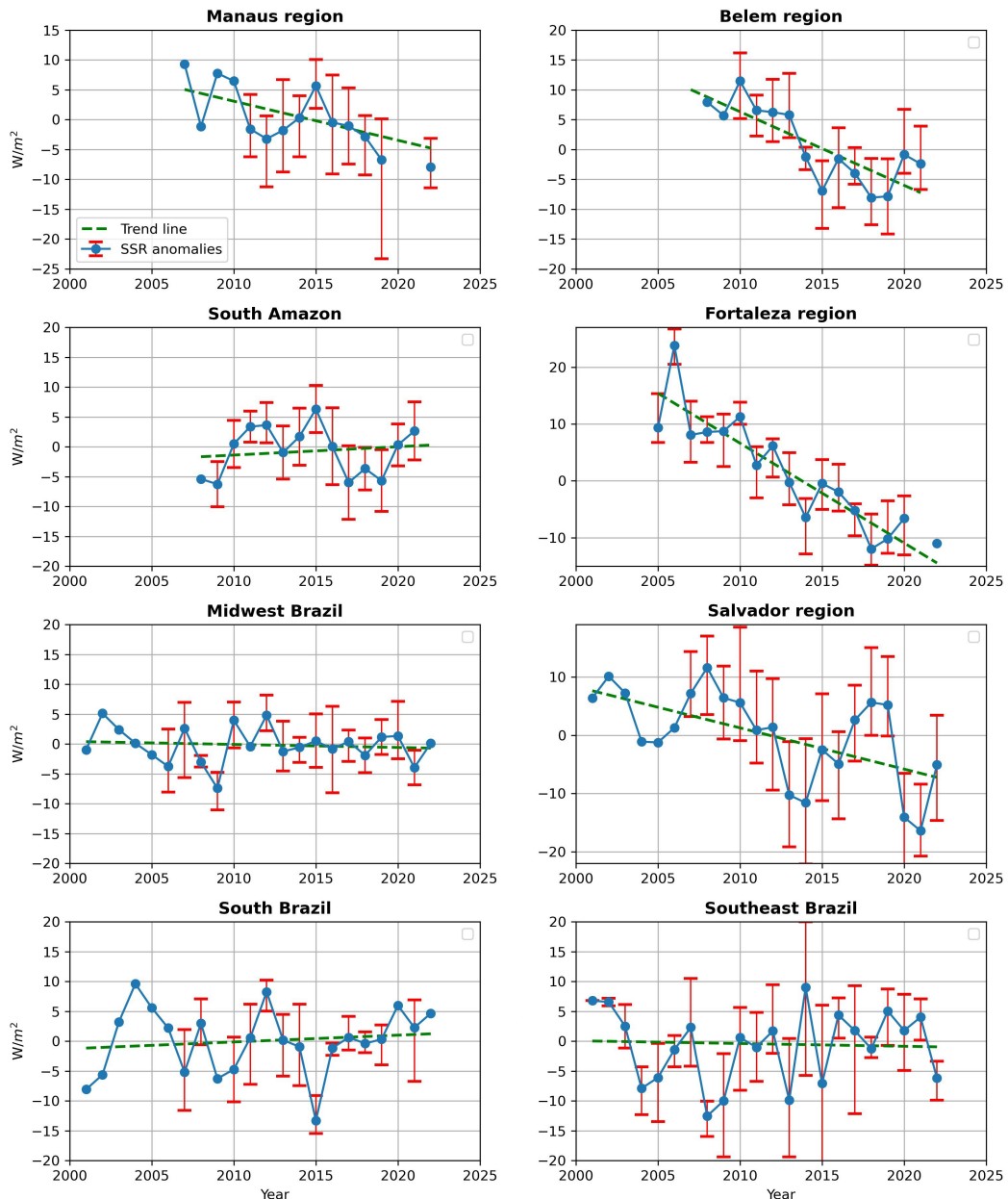


**Figure 2: Time series of all-sky Surface Solar Radiation annual anomalies from the eight composites used**
**in this study. Each composite is composed of three to five stations. In each composite, anomalies are with**
**respect to the mean of the entire period (shown in table 1). The error bars indicate the maximum and**
**minimum value for the individual stations in the respective year and composite. Trends are indicated by**
**dashed lines.**


| Composites | Period | All-sky | Clear-sky (Correa et al., 2022) | Clear-sky (Synop)* | Synop Cloud cover | All-sky atm abs | Clear-sky atm abs | CCRE |
|---|---|---|---|---|---|---|---|---|
| Manaus region | 2007-2022 | **-8.8 ± 4.2** | -2.0 ± 2.3 | - | **1.2 [0.0; 2.0]** | **0.021 ± 0.007** | 0.005 ± 0.007 | -1.1 |

| Region | Period | | | | | | | |
|---|---|---|---|---|---|---|---|---|
| Belem region | 2008-2021 | **-11.7 ± 5.8** | **-4.8 ± 2.5** | - | **1.4 [0.4; 1.3]** | **0.016 ± 0.010** | -0.001 ± 0.006 | -1.5 |
| Fortaleza region | 2005-2022 | **-16.0 ± 4.2** | **-2.7 ± 1.8** | - | 0.8 [-1.3; 2.5] | **0.034 ± 0.012** | 0.003 ± 0.011 | -0.4 |
| Salvador region | 2001-2022 | **-7.0 ± 4.5** | **-3.7 ± 1.7** | - | **1.9 [0.7; 3.1]** | **0.016 ± 0.008** | **0.010 ± 0.006** | -1.3 |
| South Amazon | 2008-2021 | 0.8 ± 6.4 | 1.6 ± 1.8 | - | - | 0.005 ± 0.015 | -0.003 ± 0.007 | - |
| Midwest Brazil | 2001-2022 | -0.4 ±2.1 | **-1.8 ± 1.1** | **-2.5 ± 1.9** | **-1.3 [-2.1; -0.3]** | **0.005 ± 0.003** | **0.005 ± 0.005** | 1.4 |
| Southeast Brazil | 2001-2022 | -0.1 ±4.5 | -1.6 ± 1.9 | -7.7 ± 8.5 | **-3.7 [-5.5; -1.3]** | 0.002 ± 0.006 | 0.006 ± 0.007 | 3.9 |
| South Brazil | 2001-2022 | 2.0 ±3.8 | 1.1 ± 4.1 | 1.2 ± 1.9 | -0.2 [-1.3; 0.7] | 0.001 ± 0.005 | -0.003 ± 0.007 | 0.2 |

**Table 1 - Trends (in W/m² per decade) for all-sky and clear-sky (using Correa et al., 2022, and using Synop cloud cover) SSR, all-sky and clear-sky (using Correa et al., 2022) fractional atmospheric absorption and Synop cloud cover at the 8 composites used in this study. Cloud Cover Radiative Effect (CCRE) referring to the Synop cloud cover trend also included - this is an estimate of the effect on SSR of the cloud cover changes. SSR trends in W/m² per decade; fractional atmospheric absorption trends in fraction (values between 0 and 1) per decade; Synop cloud cover in % per decade; and CCRE in W/m². Trends in bold are statistically significant at the 95% confidence level. Trends for Synop cloud cover were calculated using the Mann-Kendall test (see section 2.6), and as a result, the confidence interval is not always symmetrical. For this reason the confidence interval is shown in square brackets., Stations in each composite are listed in Table 3 (in appendix).**
**\*Missing values for clear-sky Synop trends occur due to the limited amount of Synop cloud cover data (0 stations for the South Amazon composite, 2 out of 4 stations for Belem and Manaus composites) or due to not enough days flagged as clear-sky in order to generate a clear-sky time series (according to the procedure described in section 2.3).**

The period covered by the data in this study should always be kept in mind, as it is shorter than long-term studies of SSR trends in regions like Europe, North America and China. However, this timespan should be enough to start identifying the relevant features affecting SSR on timescales of a decade and beyond. In the North and Northeast Brazil composites (Belem, Manaus, Fortaleza and Salvador) statistically significant (at the 95% confidence level) negative SSR trends (dimming) were observed. In the Southeast and Middle West composites, trends were negative, although near zero and statistically insignificant. Southern Amazon and South Brazil composites both show statistically insignificant positive SSR trends (brightening). This reveals a contrasting spatial distribution of the all-sky SSR trends in the first two decades of the 21th century in Brazil: while strong dimming occurred in the northern half of the Amazon region and in the northeastern coastal region, near-zero to weak positive SSR trends occurred from the southern part of the Amazon down to the south of Brazil, including the central area of the country and the densely populated southeastern region. Figure 3 shows the time series of clear-sky SSR derived with the two methods used in the study.

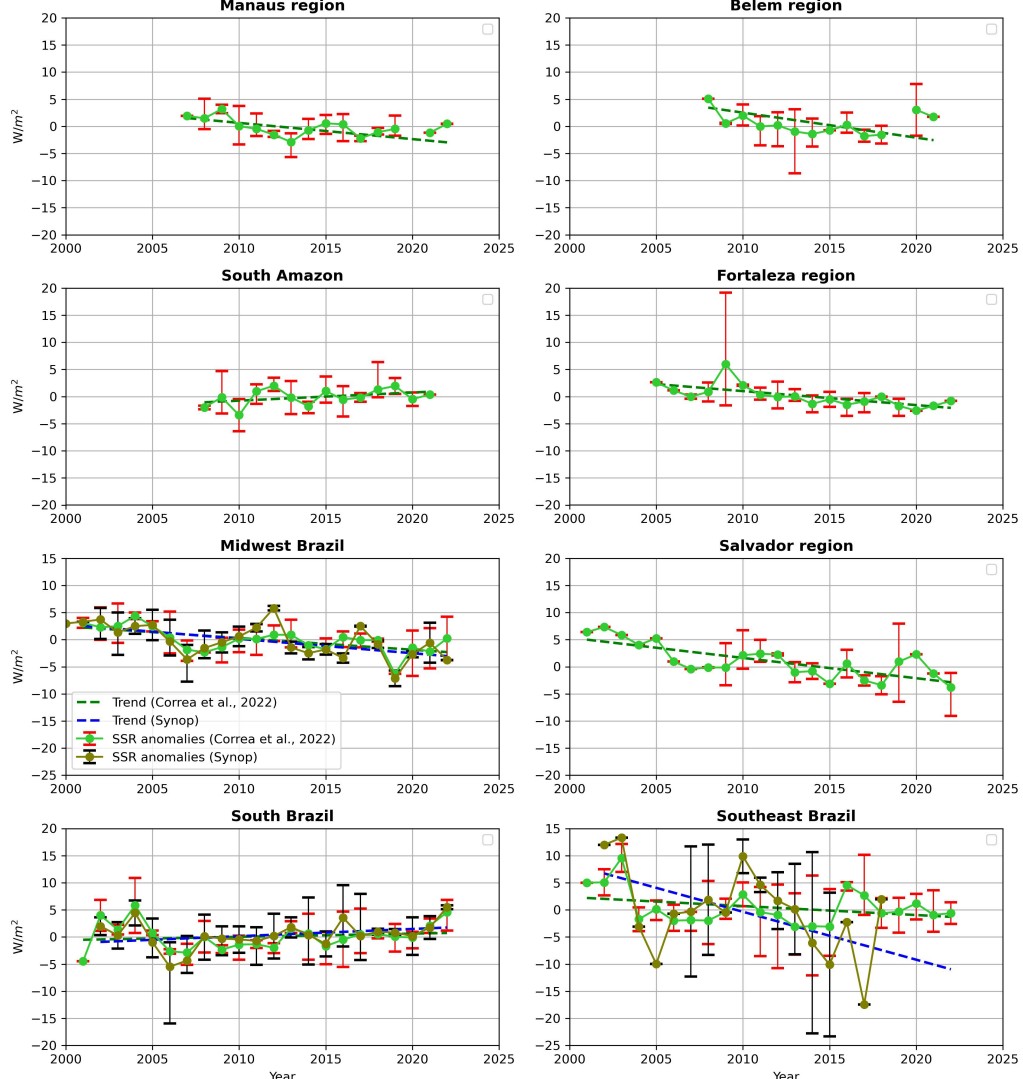

**Figure 3: Time series of clear-sky Surface Solar Radiation annual anomalies (with respect to the composite full time coverage, shown in Table 1) from the eight composites used in this study. Light green time series derived using the method by Correa et al. (2022) and olive green time series derived using Synop cloud cover to identify clear-skies. The error bars indicate the maximum and minimum value for the individual stations in the respective year and composite. Trends are indicated by dashed lines.**

Time series of clear-sky SSR based on synop cloud cover could not be derived in five out of the eight composites (see figure 3 and table 1). Synop clear-sky time series were derived when at least three stations in the composite had clear-sky data (see availability in table 3). The Manaus, Belem and South Amazon composites did not fulfill this requirement. For both Fortaleza and Salvador region composites, Synop cloud cover data was available for all stations, however, the few occurrences occurrences of low cloud cover days did not enable the derivation of clear-sky SSR time series following the procedure described in section 2.3.

Clear-sky SSR time series show in general a similar pattern as observed in all-sky. All of the composites show the same sign as the trends in all-sky, and six of them also indicate the same statistical significance (or insignificance). The only exceptions are the Midwest and Manaus composites. The former showed statistically insignificant negative trends in all-sky SSR, but statistically significant

negative clear-sky SSR trends. The opposite occurred in the Manaus composite: statistically significant
all-sky SSR trends and statistically insignificant clear-sky SSR trends. For the composites where clear-
sky data could be derived with both methods, in two of them (South and Midwest Brazil) both methods
indicate very similar inter annual variability and trends, while in the other (Southeast Brazil) the two
methods do not show strong agreement in the inter annual variability, but agreed in the direction of the
trend. Therefore, the results of the clear-sky SSR trends are supported by both clear-sky methods.
Regarding the magnitudes of the clear-sky SSR trends in comparison to the all-sky trends, another
general pattern could be observed. In all composites with statistically significant negative all-sky SSR
trends (Belem, Manaus, Fortaleza and Salvador), the clear-sky SSR trends showed a substantially
smaller magnitude. In the Southeast and Middle West, both with near-zero all-sky SSR trends, the clear-
sky SSR trends were both negative and of larger magnitude than their all-sky counterparts. In the two
composites with observed statistically insignificant all-sky SSR brightening (South Amazon and South
Brazil), the clear-sky SSR trends showed similar magnitudes as the all-sky SSR trends.
These results indicate that the clear-sky processes in the atmosphere contributed to the observed
all-sky SSR trends in the whole of Brazil, but only in the Southern Amazon and in South Brazil their
magnitude might have been large enough to be able to explain the observed SSR trends. "Clear-sky
processes" in this context refers to the interaction between solar radiation and the components of the
atmosphere without the presence of clouds. Further analysis is thus needed to better understand the
reasons for the clear-sky and all-sky decadal SSR trends observed in Brazil.
**3.2 Cloud cover, AOD and water vapour trends**
Clouds, aerosols and water vapour all can attenuate solar radiation, therefore, their variability
is analyzed in more details in this section. The order in which they are mentioned follow the order of
relevance in the discussion of solar radiation attenuation in the atmosphere, with clouds being the most
important aspect and water vapour the least important aspect. Figure 4 shows the SYNOP cloud cover
time series for 7 of the 8 composites analyzed in the study (the cloud cover time series for the Southern
Amazon composite could not be constructed due to too much missing data). The associated trends can
be found in Table 1.

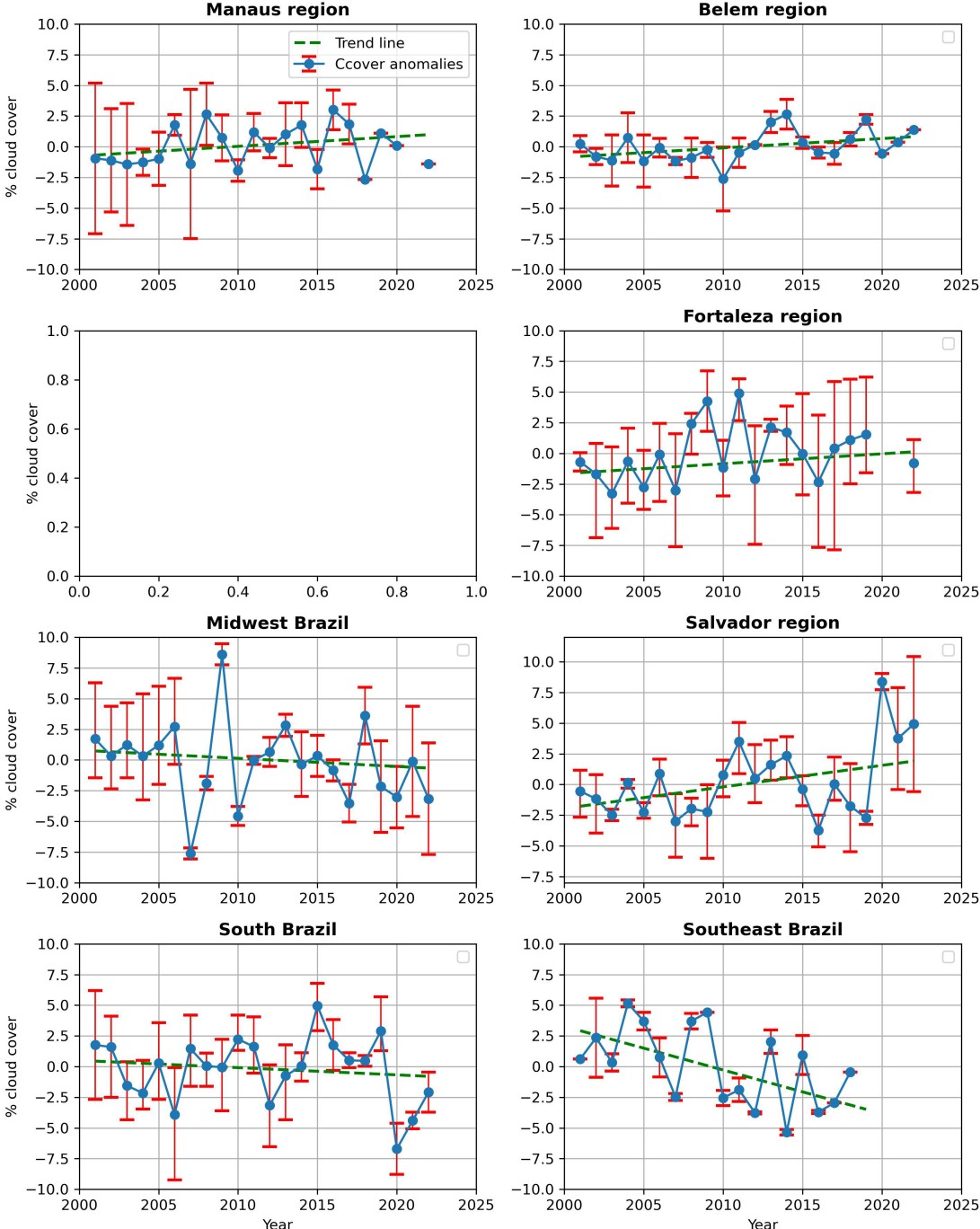

**Figure 4 - Time series of annual mean Synop cloud cover anomalies for seven of the eight composites used in this study. Not enough data was available to derive a time series for the South Amazon composite. The error bars indicate the maximum and minimum value for the individual stations in the respective year and composite. Trends are indicated by dashed lines.**

The Manaus, Belem and Southeast Brazil composites do not have synop cloud cover data for all stations (2 out of 4 available for Manaus and Belem, and 3 out of 5 for Southeast Brazil), therefore a comparison between all-sky SSR trends and Synop cloud cover at these composites is based on the assumption that the cloud cover observations for the composites are representative for all stations. This is a reasonable assumption given the geographical proximity between the stations within these three composites and the lack of any climatic or geographical feature that can strongly affect cloudiness at

individual stations (e.g. high topography). In these composites, all stations are located in areas with the
same precipitation regimes as classified by Ferreira and Reboita (2022), also corroborating with the
assumption of good representativeness.

Cloud cover trends are in most cases consistent in sign with the all-sky SSR trends. That is,

positive (negative) trends in cloud cover occurring during a period of negative (positive) SSR trends.
That is the case for the four composites with statistically significant all-sky SSR dimming (Belem,
Manaus, Fortaleza and Salvador). They all show positive trends in cloud cover, and all, except
Fortaleza, show statistical significance. This is consistent in the sense that the increase in cloud cover
contributes to the observed decrease in SSR, especially considering that the magnitude of the clear-sky
SSR trends at these locations was significantly smaller than the all-sky SSR trends. However,
quantitatively, the small magnitude of the cloud cover trends (between 0.8 and 1.9 % per decade)
challenges any hypothesis of a major contribution of cloud cover changes to the decadal SSR trends.
That is, the cloud cover trends are too small and, and as a consequence, the contribution of changes in
cloud cover to the SSR trends is expected to be minor. This contribution is estimated objectively by the
CCRE (see table 1), which shows, in most cases, low values (in comparison to the all-sky SSR trends),
suggesting only a minor contribution from cloudiness to the SSR trends.

Cloud cover trends show near-zero values in the South region, suggesting no major cloud cover

contribution to the SSR trends. Southeast and Middle West show both statistically significant negative
trends in cloud cover, with remarkably strong values in SE (-3.7 [-5.5; -1.3] % per decade). Both
composites show near-zero but negative all-sky SSR trends, with stronger negative clear-sky SSR
trends. Thus, the cloud cover trends exert an opposite effect to the one of the clear-sky processes at both
composites. This is also consistent, in the sense that with clear-sky processes and cloud cover having
competing opposite effects, if their magnitude is similar, their effects cancel out, and the resulting all-
sky SSR trend would be near zero.

Figure 5 shows the decadal trend maps of annual AOD in the 2003-2020 period from CAMS

reanalysis and of total column water vapour in the 2001-2020 period from ERA5.

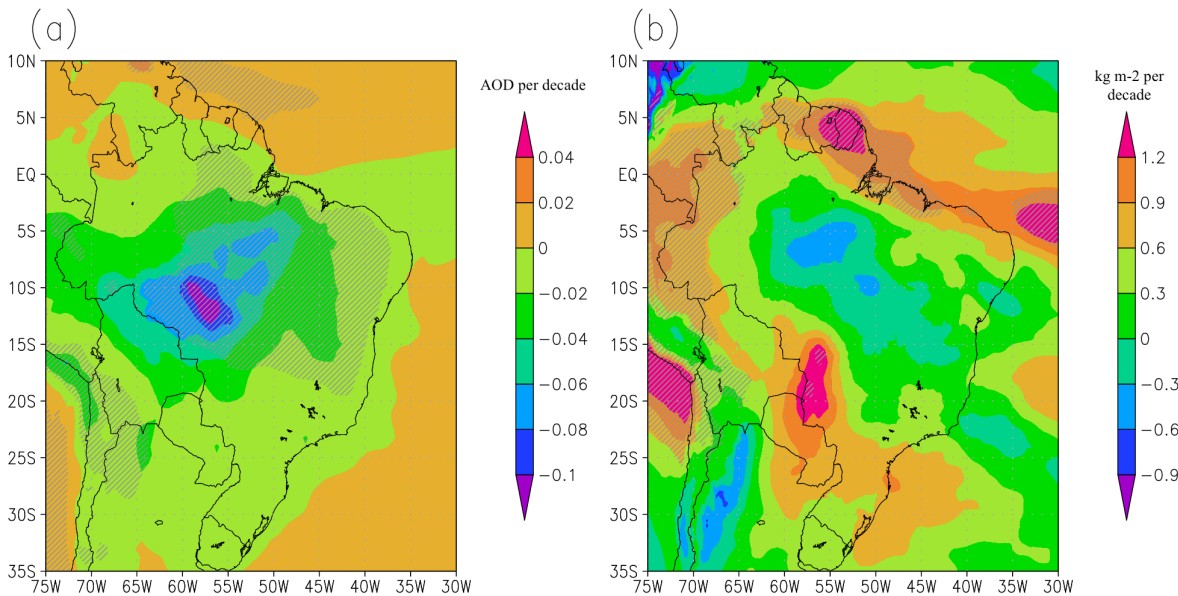

**Figure 5 - Maps of decadal trends of (a) AOD [unitless] in the 2003-2020\* period from CAMS reanalysis**
**and of (b) total column water vapour [in kg m$^{-2}$] in the 2001-2020 period from ERA5. Shaded areas indicate**
**statistical significance (at the 95% confidence level).**
**\*The dataset was available only from 2003 onwards.**

In Figure 5 (a) we see a strong negative AOD trend in the Southern Amazon in the period, and

slightly negative to near-zero trends in the rest of the country. Trends are statistically significant in the
Southern Amazon and in the inner area of the country down to approximately 15 degrees south (shaded
areas), while they lose significance towards the coast. The process that dominates the AOD trends in
the Southern Amazon (and in the whole country) is the reduction of biomass burning in the Amazon
region. The Southern part of the Amazon region is the area that suffers most from the biomass burning
(Artaxo et al., 2006), especially in the dry season during the southern hemisphere winter. A reduction
in forest fires at the beginning of the 21st century has been reported (Silva Junior et al., 2021), and its
effect is clear in the AOD trends. This result is consistent with the observed clear-sky SSR brightening
in the Southern Amazon, but challenges the negative clear-sky SSR trends observed in most of the
country. This suggests that changes in AOD were not primarily responsible for the clear-sky SSR trends
in the whole of Brazil, with the exception of the Southern Amazon region.

The water vapour trend map (Figure 5 (b)) shows remarkably negative trends in the central

Amazon, in a region around the east coast of Brazil and in the southernmost part of the country.
Remarkably positive trends are present from the middle Western Brazil (south of the Amazon region)

stretching to Southeastern Brazil, and in the northeast and north coastal regions of the country. The spatial distribution of the decadal variability of water vapour does not generally comply with the observed clear-sky SSR trends. We used these trends to estimate the change in atmospheric clear-sky absorption due to solely water vapour, using the empirical model presented by Hakuba et al. (2016). Based on these estimations, even in a region with strong water vapour trend such as Midwest Brazil, these changes would be responsible for an increase in atmospheric clear-sky absorption (and consequently decrease in SSR) of approximately 0.4 $W/m^2$ per decade. This is almost one order of magnitude smaller than the clear-sky SSR trends in the region (-1.8 and -2.5 W/m2 per decade, for clear-sky conditions based on Correa et al. (2022) and Synop cloud cover, respectively). This suggests that the water vapour contribution to the observed clear-sky SSR trends, when existed, was only minor.

### 3.3 Atmospheric absorption and Anthropogenic emissions

To better understand the reasons behind the observed clear-sky SSR trends and the overall processes responsible for the all-sky SSR trends, we analyzed the changes in fractional atmospheric absorption under all-sky and clear-sky conditions. This is a relevant aspect to be assessed, because changes in atmospheric shortwave absorption can be an important driver of SSR trends (Schwarz et al., 2020). Figure 6 shows these time series for the composites considered in this study both under all-sky and clear-sky conditions.

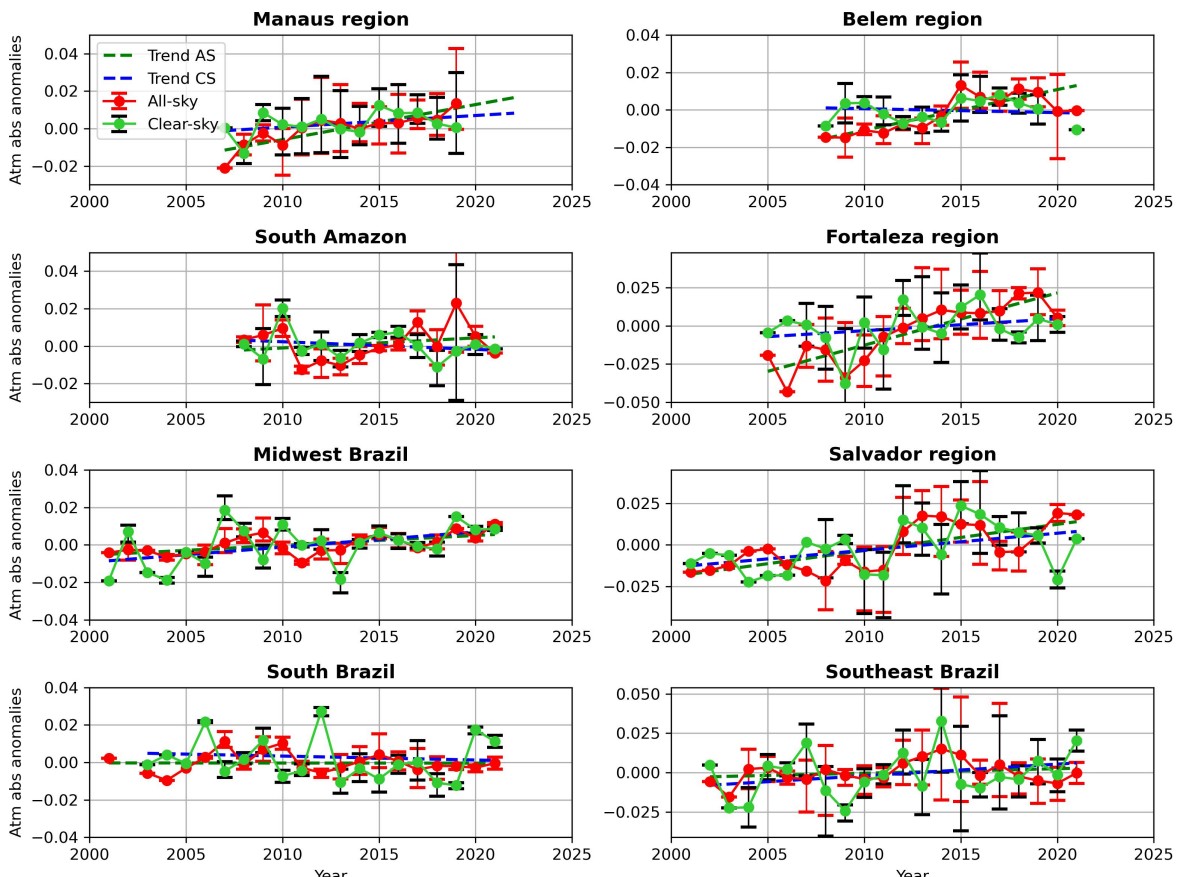

**Figure 6: Time series of all-sky (red) and clear-sky (green) fractional atmospheric column absorption annual anomalies for the eight composites used in this study.**

The fractional atmospheric absorption under all-sky conditions increased in most of the composites in the first two decades of the 21st century. Five composites showed statistically significant positive trends in $F_{abs}$, they are: Manaus, Belem, Fortaleza, Salvador and Middle West. The other composites also showed positive trends, but they were statistically insignificant. Under clear-sky conditions, the trends are obviously smaller, as the cloud induced multiple scattering does not play a role in enhancing column absorption. Only in the Salvador and Middle West composites statistically significant positive trends were observed in the atmospheric absorption under clear-sky conditions. All the other composites show statistically insignificant trends with the same sign as their all-sky counterparts, with the exception of the South Amazon, which shows statistically insignificant negative trends under clear-sky, contrasting to a statistically insignificant positive trend under all-sky conditions.

These results reveal two important aspects of the SSR variability in Brazil. First, in seven out of the eight composites the changes in clear-sky absorption comply with the clear-sky SSR trends. That is, increasing (decreasing) clear-sky atmospheric absorption was always linked to a decrease (increase) in clear-sky SSR. Secondly, the presence of clouds greatly increased atmospheric absorption (not shown) but also its trends. This has most likely happened because of the intensification of multiple scattering occurring under partially cloudy skies, resulting in a magnification of the trends seen in clear-sky conditions. This is reinforced by the fact that the strongest all-sky atmospheric absorption trends were found in the four cloudiest composites (Manaus, Belem, Fortaleza and Salvador), which happen to be the four composites with statistically significant negative all-sky SSR trends (dimming). Even though these results are consistent with each other, they also suggest that AOD only showed strong trends in the South Amazon region, and that water vapour only contributed as a minor part to the observed changes in atmospheric absorption (see discussion above). Thus, this raises the question of what could be the main responsible for the changes in atmospheric absorption in Brazil. To try to answer this question, we analyzed the decadal trend in aerosol absorption optical depth (AAOD) at 500 nm from OMI. The trend map is displayed in Figure 7.

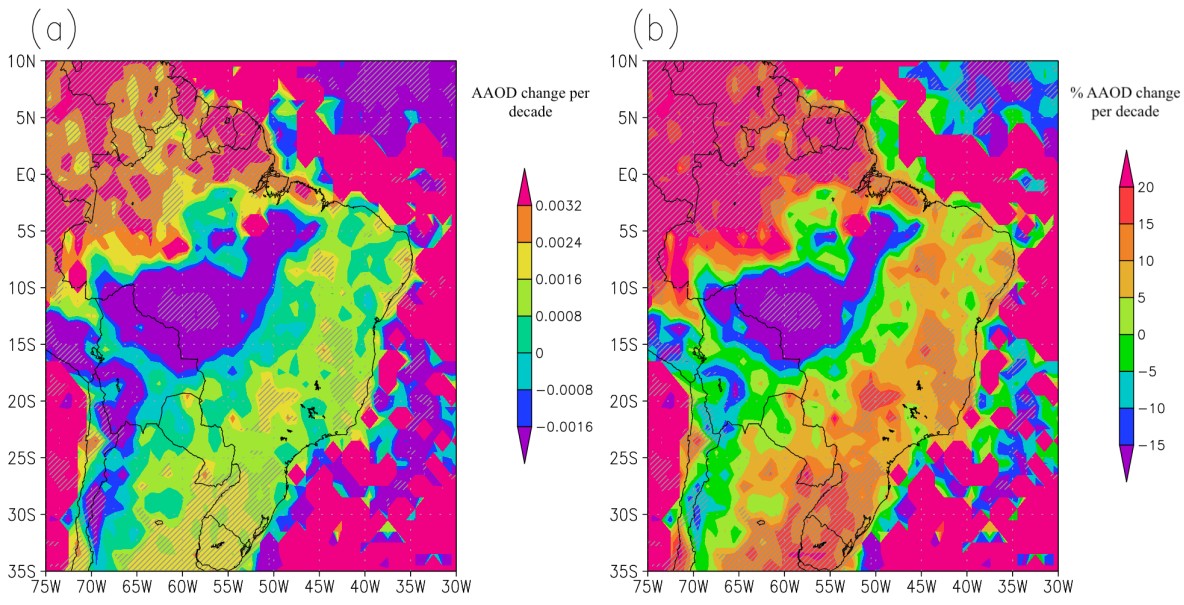

**Figure 7: (a) Absolute and (b) relative (%) decadal trends in Absorption Aerosol Optical Depth (AAOD)**
**in the 2005-2022 period from OMI. Shaded areas indicate statistical significance (at the 95% confidence**
**level).**

The map shows a clear distinction between the region under strong influence of the forest fires

in the Amazon (South Amazon) and the rest of Brazil. In the South Amazon, the data shows a decrease
in AAOD in the 2005-2022 period, while in the rest of the country an increase in absorption AAOD at
500 nm is observed. The spatial distribution of the trends suggests that the reduction in AAOD in the
South Amazon could be associated with the forest fires reduction also visible in the AOD trends. In the
whole rest of the country, positive trends in AAOD are observed. This reveals a significant change in
the optical properties of the aerosols present in Brazil in the first two decades of the 21st century, with
a trend towards more absorbing aerosols (at 500 nm) in most of the country. The AOD trend map
(Figure 5a) shows that in the same areas where AAOD increases, AOD remains nearly constant, with
trends close to zero. In order to better visualize potential reasons for an increase in AAOD at 500 nm
in most of Brazil, we also investigated trends in anthropogenic $SO_2$ and Black Carbon emissions in
Brazil. They are displayed in Figure 8.

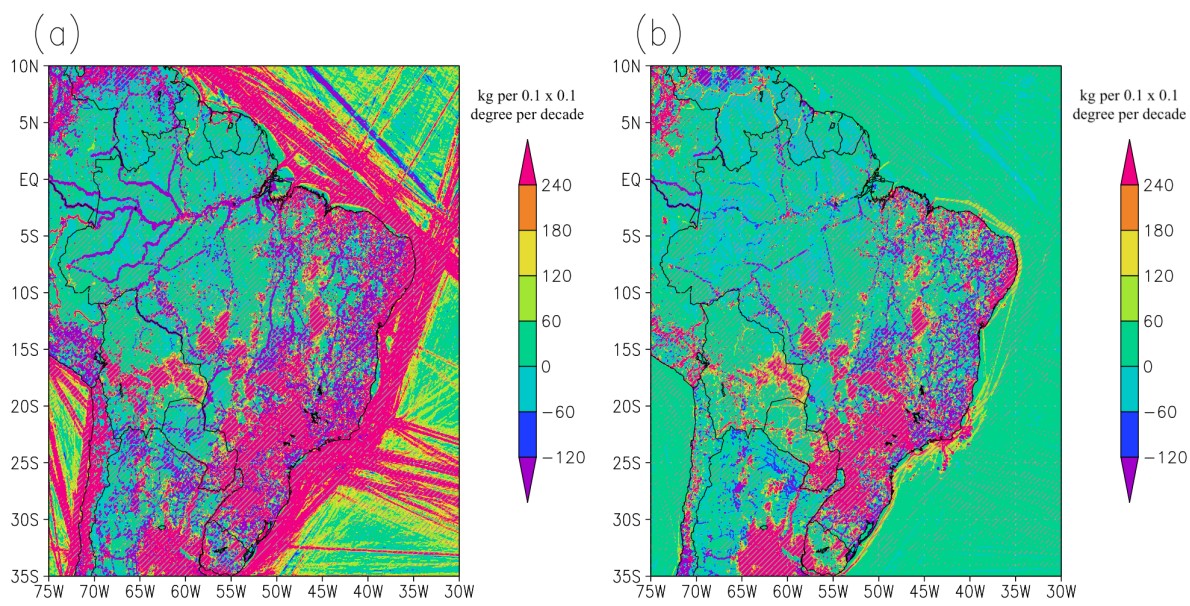

**Figure 8: Decadal trends of annual mean (a) SO₂ and (b) Black Carbon anthropogenic emissions (in kg per**
**0.1x0.1 degree grid per decade) for the 2001-2018 period from EDGAR. Shaded areas indicate statistical**
**significance (at the 95% confidence level).**
They show a general increase in anthropogenic emissions in most of Brazil, especially in highly
populated areas. The only areas not showing increase in anthropogenic emissions are in the Amazon
rainforest. This might be counterintuitive when comparing the emissions trends (figure 8) with AOD
trends (figure 5a), as the strongest AOD trend is observed in the south of the Amazon region. However,
EDGAR emission estimates do not consider large scale biomass burning, land use change and forestry
(Crippa et al., 2018). As discussed in section 3.2, the AOD negative trend is mostly associated with
reductions in biomass burning in the first two decades of the 21st century in the Amazon. Therefore,
the biggest cause of the AOD trend (Figure 5a) is not considered in the emission data used in Figure 8.
Even though, according to figure 8, anthropogenic emissions did not increase significantly in
the Amazon region, emissions still increased around the biggest cities in the region, like Manaus and
Belem. This is of special relevance for this study, since seven of the eight composites are centered
around cities with over a million inhabitants, where the large and usually increasing population (Lobo
and Cunha, 2019) plays an important role to the atmospheric composition. The only composite that
does not follow this rule is the South Amazon composite, where the biggest city is Porto Velho, which
in 2020 had a population of less than 500'000 people (IBGE 2022). As anthropogenically emitted
aerosols tend to account for a larger fraction of solar radiation absorption than natural aerosols (Wang
et al., 2009), this increase in anthropogenic emissions (especially of black carbon) complies with the
increasing AAOD in most of Brazil. Even though sulphate aerosols absorb much less shortwave
radiation than black carbon, the increasing presence of scattering aerosols can also have a similar effect
to the presence of broken clouds for atmospheric absorption (as discussed for the composites in North
and Northeast Brazil): they increase multiple scattering, increasing the optical path of the photons,
which increases the chances for absorption by the atmosphere. Therefore, the increasing anthropogenic
emissions complies with the observed increase in atmospheric absorption in most of Brazil in the period
of study. Similar results indicating a stronger impact of the changes in optical properties of the aerosols
than the changes in aerosol optical depth on the observed SSR trends were also found for Japan in the
1990s by Kudo et al. (2012).
**4.    Discussion**
**4.1 Physical consistency of the results**
The results of this study point to a relevant impact of changes in atmospheric absorption in at
least half of the regions analyzed. However, this is based on the fractional atmospheric absorption data,
which is derived (as described in section 2.4) by combining in situ SSR (point) measurements with
gridded data of surface albedo and outgoing shortwave radiation at TOA, at 0.25 and 1.0 degree spatial
resolution, respectively. So the first question to be addressed is whether these results can be trusted
even with the use of different spatial resolutions. Schwarz et al. (2018) investigated the spatial
representativeness of SSR measurements in many stations around the world, including four stations in
Brazil: Florianopolis, São Martinho da Serra (both in South Brazil), Brasilia (in Middle West) and
Petrolina (~ 450 km away from Salvador). The authors found a good representativity of SSR for the 1-
degree surroundings at most stations around the world at the monthly time scales, with estimated
decorrelation lengths (the distance over which a point measurement is representative) always higher
than 3 degrees in all of the four Brazilian locations. Madhavan et al. (2017) investigated spatial
representativeness of SSR measurements at shorter time scales, and found that point measurements
were representative to a 10 x 10 km area in time scales up to around one hour (from 26 minutes at
overcast conditions to 70 minutes at broken clouds conditions). The authors demonstrated that the
decorrelation lengths increase linearly (on a log-log scale) with decreasing frequency (longer time
averaging). Following the results of the study by Madhavan et al. (2017), this would lead to
decorrelation lengths around the order of 100 km (~1 degree) at the daily (24-hour) time scales.
Therefore, based on the interpretation of these results, we can expect a satisfactory consistency in the
results from combining point measurements at the surface with 1-degree measurements at the TOA at
daily time scales, as done in this study. An in-depth analysis to estimate the decorrelation lengths at
daily time scales of each station goes beyond the scope of the study.
The performance of the gridded products used used in this study are discussed in their
respective documentations, referenced in section 2. Spectral surface albedo is reported as a main source
of uncertainty in the satellite based products, especially OMI AAOD, however, this tends to be a major
problem over the ocean. Sub-grid cloud contamination tends to also represent a problem for the retrieval
of satellite based products. But this is reported to lead to an over/under estimation of the average AAOD,
but should not affect the representation of its long-term variability. No issues with the long-term
variability of the reanalysis products were reported.
Regarding atmospheric absorption, previous studies (e.g. Li et al., 1995; Byrne et al., 1996)
have shown an enhancement in atmospheric absorption under cloudy conditions. According to previous
literature, such an enhancement would not be caused by cloud absorption, but by cloud scattering, which
increases the optical path of a photon in the atmosphere, consequently increasing the chances of this
photon to be absorbed by other components of the atmosphere, such as water vapour and aerosols. Even
though the existence of this mechanism is clear, the quantitative influence this could have on the energy
budget at any location would also depend on the characteristics of cloud occurrence (e.g. the frequency
of cloud free, overcast and partially cloudy conditions). As much as cloud free conditions are not
optimal for atmospheric absorption, completely overcast conditions are not either. Under fully cloudy
conditions, the backscattering of incoming shortwave radiation is high, usually not increasing the
optical path of the photons and not allowing them to reach lower levels of the atmosphere, where water
vapour and aerosol concentrations are higher. Thus, the high occurrence of partially cloudy conditions
would increase the cloud effects on atmospheric absorption, via the increase in the optical path of the
photons. Such conditions are found in Belem, Manaus, Fortaleza and Salvador, due to the importance
of local convection for cloud formation in such hot and humid locations. The differences in the
fractional atmospheric absorption trends between clear-sky and all-sky conditions at these locations
reinforces this: trends under all-sky conditions are one order of magnitude larger than their clear-sky
counterparts. This is not observed at all the other locations, which have a higher dependence on
mesoscale and synoptic scale phenomena for cloud formation than the previously mentioned locations.
In fact, a difference in the precipitation regimes between the region where all the four above-mentioned
composites are located and the rest of Brazil has already been pointed out by Reboita et al. (2010) and
by Ferreira and Reboita (2022).
Chtirkova et al. (2023) investigated the potential effect of internal variability on the SSR trends,
and the relevance especially of Atlantic oceanic modes like the Atlantic Meridional Mode (AMM) or
the Atlantic Multidecadal Oscillation (AMO) to affect SSR trends by changing cloudiness in Brazil.
The AMM and AMO went to lower values during the period of study (2001-2022), which should lead
to decreasing SSR in Northeastern Brazil. This is consistent with the negative SSR trends in the region.
But it is important to note that this reduction in the oceanic modes values did not represent a major
phase transitions of these modes. A major increase in AMO occurred in the 1990s, and the cloud cover
trends (from ERA5) for the 1990-2006 period show a strong decrease in cloud cover in most of Brazil,
especially the south and western part of the country. No SSR data was available for further investigation
in this study, but the importance of internal variability for SSR trends should not be neglected in future
studies

The trends in SSR and supporting information in the eight composites made it possible to

separate the discussion of the causes for the SSR trends into three groups. The composites in each group
and their common characteristics are listed on table 2.

| Composites | Common characteristics |
|---|---|
| Manaus, Belem, Fortaleza and Salvador | Strong all-sky dimming |
|  | Distinguished clear-sky dimming with lower magnitude than the all-sky |
|  | Positive cloud cover trends |
|  | Positive trends in all-sky atmospheric absorption |
|  | Positive trends in clear-sky atmospheric absorption, but one order of magnitude smaller than their all-sky counterparts (this item does not apply to Salvador) |
| Southeast Brazil and Midwest Brazil | Strong negative cloud cover trends |
|  | Negative clear-sky SSR trends |
|  | Negative and statistically insignificant SSR trends |
| South Amazon and South Brazil | Statistically insignificant all-sky brightening |
|  | Statistically insignificant clear-sky brightening |


**Table 2: Groups of composites with and their common characteristics as indicated by the results**
**presented in this study.**

Based on this, we separated the discussion on the causes for SSR trends in three sections, each

discussing one of the three groups.

### 4.2 Dimming in North and Northeastern Brazil

In this section, we discuss the dimming observed in the Manaus, Belem, Fortaleza and Salvador

composites, located in North and Northeastern Brazil. All of the composites showed statistically
significant all-sky dimming in the period, associated with a clear-sky dimming which was statistically
significant in all composites, except Manaus. The difference in the magnitude of the all-sky SSR trends
(from -6.3 $W/m^2$ per decade in Salvador to -18.8 $W/m^2$ per decade in Fortaleza) to the clear-sky SSR
trends (from -2.0 $W/m^2$ per decade in Manaus to –4.8 $W/m^2$ per decade in Belem) in the four composites
suggests that the clear-sky processes alone are unlikely to be strong enough to explain the SSR trends
in these locations. However, the fact that the clear-sky trends show the same sign as the all-sky trends,
with (in most cases) statistical significance, indicates that processes occurring under clear-skies did
contribute significantly to the overall trends. The contrast between all-sky and clear-sky also indicates
a potential contribution of changes in cloud cover to the trends. In fact, we identified positive cloud
cover trends, consistent with the observed reduction in SSR, but the magnitude of the trends (from 0.8
% per decade in Fortaleza to 1.9 % per decade in Salvador) and the resulting impact of these cloud
cover trends on the SSR trends, estimated by the CCRE (see Table 1), is small when compared to the
SSR trends. Thus, our results (summarised in the table 1) suggest contributions from both clear-sky
processes and cloud cover to the SSR trends, but none of them show a remarkable dominance compared
to the other.
Further analysis of the atmospheric absorption showed strong positive (and statistically
significant) trends in atmospheric absorption in all four composites. Schwarz et al. (2020) have shown
that changes in atmospheric shortwave absorption can be an important driver of dimming and
brightening. We also found that the atmospheric absorption trends were greatly enhanced by the
presence of clouds. This happens because the scattering by clouds increases the optical path of the
photons. This effect occurs primarily under broken clouds conditions, when three-dimensional multiple
scattering magnifies this effect. Our findings comply with the results presented by Byrne et al. (1996)
and references therein, which highlight the enhancement of atmospheric absorption of solar radiation
under broken clouds conditions. Results from Li et al. (1995) also suggested that this effect is stronger
in tropical regions, and the authors discuss that this is associated primarily aerosol and water vapour
absorption rather than cloud absorption. The characteristics of the distribution of cloudiness in the four
composites, displayed in Figure 9, might also play a role in this process. Stations from these composites
tend to have frequent occurrences of partially cloudy conditions. In the Belem, Fortaleza and Salvador
composites the daily cloud cover is between 25% and 80% in around two thirds of the days. For the
Manaus and Midwest Brazil composites this range of cloud cover occurred in around half of the days
and for South and Southeast Brazil composites this value is around one third. Thus, at the daily scale
we see a dominance of partially cloudy occurrences at three out of the four composites discussed in this
section. Even though the same distinguishable characteristic was not found for the Manaus composite
at the daily scale, based on the regionalization of precipitation regimes by Reboita et al. (2010), we
would expect the same finding at a more refined time scale also for the Manaus composite. That would
be the expectation because of the higher relevance of local convection at hot and humid locations
(convective clouds cause more broken cloud fields than large scale synoptic clouds) at the four
composites discussed in this section, in comparison to the other composites, where mesoscale and
synoptic meteorological systems tend to play a more important role for cloud formation. This higher
occurrence of broken clouds in the regions of the four composites discussed in this section then tends
to play an important role for the enhancement of atmospheric absorption.

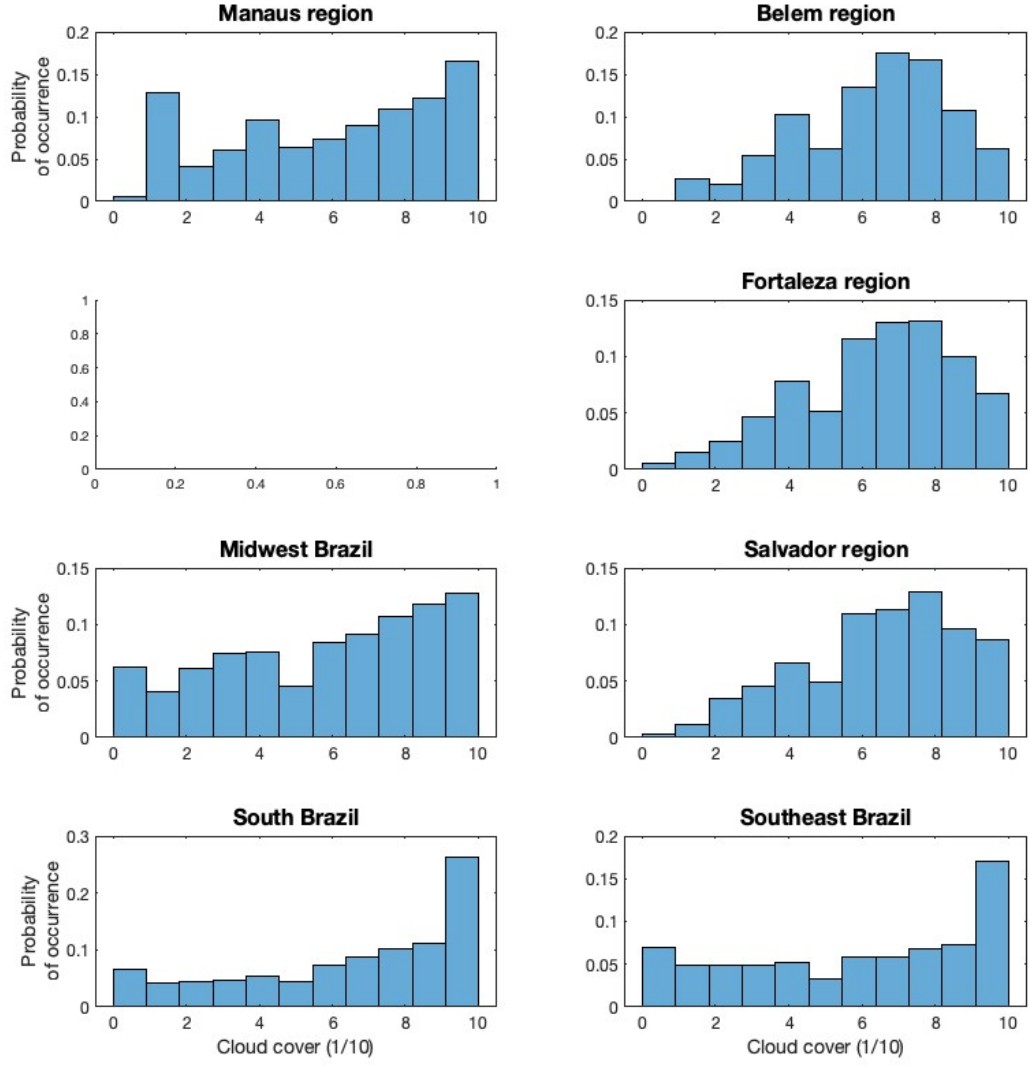

**Figure 9: Distribution of daily Synop cloud cover occurrences in seven out of the eight composites used in this study. Not enough data was available to derive a distribution for the South Amazon composite.**

A simple multiplication between the incoming TOA radiation at each composite and the trends in fractional all-sky atmospheric absorption (shown in Table 1) reveals an estimated increase in all-sky atmospheric absorption from approximately $6 \pm 3$ W/m$^2$ per decade in Belem up to $14 \pm 5$ W/m$^2$ per decade in Fortaleza. If we assume that such an increase in atmospheric absorption is directly reflected in a reduction in SSR, we find that the effect of changes in atmospheric absorption under all-sky conditions have a higher effect than the estimated clear-sky SSR trends (see table 1) and the estimated effects of changes in clouds cover (see CCRE in table 1), and are more consistent with the magnitude of the observed all-sky SSR trends (presented in table 1). Thus, these results suggest that the increase in atmospheric absorption was the strongest contributor for the negative SSR trends observed in these four composites in north and northeast Brazil, with contributions also from changes in cloud cover. The difference in the all-sky and clear-sky absorption trends at these four composites indicates that clouds played an important role in the increasing in absorption, most likely by enhancing the optical path of

photons via multiple scattering under partially cloudy conditions. The results also suggest that these
changes in atmospheric absorption were greatly influenced by the changes in the optical properties of
the aerosols present in these regions. Our results showed the occurrence of increasing anthropogenic
emissions of $SO_2$ and black carbon, which did not seem to significantly change the AOD (possibly
because of its competing effects with the reduction of biomass burning emissions in South Amazon),
but increased the AAOD. This is most likely the cause for the increase in atmospheric absorption at the
four composites. All of this points to a relevant influence of anthropogenic factors to the SSR trends in
the first two decades of the 21st century in the regions around Manaus, Belem, Fortaleza and Salvador.
Remembering that these are all big cities with over a million inhabitants each, therefore this result could
be biased towards big population centers.
**4.3 Midwest and Southeast Brazil**
In this section we discuss the causes of the decadal SSR trends in the Middle West and
Southeast Brazil composites. Both all-sky SSR composites show near-zero trends, with -0.4 ± 2.7 W/m$^2$
per decade in the Middle West and -0.6 ± 5.4 W/m$^2$ per decade in Southeast Brazil in the first two
decades of the 21st century. Both composites show clear-sky SSR dimming (statistically significant in
the Middle West and statistically insignificant in the Southeast) and statistically significant decrease in
cloud cover in the period. An increase in atmospheric absorption was also observed at these locations,
but the trends were substantially smaller than the trends observed in the four composites discussed in
the previous section. These results already suggest different physical processes playing a role in the
causes of SSR decadal trends in these regions.
The trends in fractional clear-sky atmospheric absorption in the two composites are similar to
each other (0.0051 ± 0.005 per decade in the Middle West and 0.0059 ± 0.007 per decade in the
Southeast) and are larger than the trends in three out of the four composites discussed in the previous
section. The clear-sky absorption trends are also larger than the all-sky absorption trends in the Middle
West and Southeast. This indicates a bigger relative relevance of the cloud-free processes for the SSR
trends in these two regions compared to the four locations previously discussed. This is reinforced by
the clear-sky SSR dimming at the two locations, and is also most likely associated with increasing
anthropogenic emissions, which lead to more absorptive aerosols, without significant change in AOD.
A comparison between the results of these two composites with the four composites in North
and Northeast Brazil supports the discussion regarding the impacts of broken clouds to the solar
atmospheric absorption and the distribution of cloud cover occurrences, presented in the previous
section. As discussed by Reboita et al. (2010) and by Ferreira and Reboita (2022), in the region from
Middle West to Southeastern Brazil a stronger influence of large scale synoptic meteorological systems
like cold fronts, South Atlantic Convergence Zone (SACZ) and the South American Low Level Jet
(SALLJ) contrasts with Northern and Northeastern Brazil, where local convection and circulation play
a more important role. This leads to different precipitation and cloudiness regimes between the
composites discussed in this section and in previous section. These regimes magnify the effects of
atmospheric absorption in the North and Northeastern Brazil, again fitting to the results by Li et al.
(1995), while not doing so in the rest of the country.

The results of these two composites also show a significant positive effect of changes in cloud
cover on the SSR trends. Strong significant negative trends in cloud cover were observed at both
regions. As a result of the competing effects between cloud-free processes and changes in cloud cover,
the resulting SSR trends in the first two decades of the 21st century were negative, but near-zero, for
both composites. This shows opposing effects of anthropogenic (changes in aerosols) and natural
(changes in cloud cover) changes canceling out.

**4.4 South Amazon and South Brazil**

In this section we discuss the causes for the SSR decadal trends in the South Amazon and South
Brazil. In both regions statistically insignificant brightening was observed in the all-sky SSR trends.
Clear-sky SSR trends also showed brightening (statistically insignificant) in both regions. Cloud cover
trends in South Brazil were rather small (-0.4 [-1.4; 0.6] % per decade), while cloud data was not
available for South Amazon.

For the South Amazon, the most relevant aspect to be discussed is the strong negative trend in
AOD observed in the study period, associated with the documented reduction in deforestation and
biomass burning in the Amazon (Silva Junior et al., 2021). Amazon biomass burning aerosols play an
important role in the atmospheric transmissivity in the region, but their emission, and consequently their
effects, are highly seasonally dependent, as shown by Schwarz et al. (2019). For this reason, even
though the annual AOD decadal trends show very strong negative values, the strong effects on SSR are
present mostly in the dry season (southern hemisphere winter), and are smoothed out with annual means
and decadal trends calculations. The seasonal clear-sky SSR trends in this composite are positive
(statistically insignificant at the 95% confidence level) in winter and spring (5.0 $\pm$ 5.6 and 1.1 $\pm$ 3.9
W/m$^2$ per decade, respectively) and negative (statistically insignificant at the 95% confidence level) in
summer and fall (-2.6 $\pm$ 2.7 and -1.6 $\pm$ 3.3 W/m$^2$ per decade, respectively), reinforcing this hypothesis.
This smoothing of the AOD effects in the annual means and decadal trends is most likely the reason
why, despite the strong negative AOD trends in the region, the all-sky and clear-sky SSR trends show
positive trends with an absolute magnitude remarkably smaller than the trends observed in north and
northeastern Brazil. This counterintuitive result (strong negative AOD decadal trend not resulting in
strong brightening neither in all-sky nor in clear-sky SSR) reveals the importance of taking seasonality
into account when investigating the response of SSR to changes in AOD.

In South Brazil, the SSR decadal trends are weakly positive, not of statistical significance, both
under all-sky and clear-sky conditions. This suggests the lack of a strong driver for the SSR trends in
the period analyzed. Cloud cover shows a small negative trend (statistically insignificant). Near-zero
trends are also found in AOD and in atmospheric absorption. The map of AAOD at 500 nm shows small

positive trends in the period, but water vapour shows small negative trends. It is important to note that due to the logarithmic response of atmospheric absorption to changes in water vapour (e.g. Hakuba et al., 2016), this is the region in Brazil with the expected strongest sensitivity to changes in water vapour. Combining all these results together denotes competing small effects from different sources, and this is most likely the reason for the resulting non significant trend observed. Another relevant aspect to be highlighted, is that the period of analysis did not show a strong transition in the signal from oceanic modes in the Atlantic. Chtirkova et al. (2023) pointed out the importance of the AMM and AMO oceanic modes for the SSR trends in South America. This could be relevant for all composites, but the lack of strong effects on SSR changes of the existing forcing elements in South Brazil in the post 2000 period let us to hypothesize that in a transitional period of AMM and/or AMO, internal variability could dominate the SSR trends in this region, especially via changes in cloud cover. This hypothesis is reinforced by the cloud cover trends from ERA5 for the 1990-2006 period (Figure A1, in appendix), which show strong negative cloud cover trends in the region associated with the transitioning of the AMO from a negative to a positive phase. The expectation is that the cloud cover trends in this period dominated the SSR trends, causing brightening in the last decade of the 20st century in South Brazil. However, the lack of SSR data before 2000 did not allow us to verify this hypothesis.

## 5. Conclusions

In this study we presented and investigated the magnitudes of the SSR trends and their associated causes over the first two decades of the 21st century based on 34 stations in Brazil, divided into 8 composites of 3 to 5 stations each. These are: Manaus region, Belem region, South Amazon, Fortaleza region, Middle west, Salvador region, Southeast Brazil and South Brazil. The exact temporal coverage of the SSR time series was composite-dependent, covering 22 years (2001-2022) in the four southernmost composites (South, Southeast, Middle West and Salvador), and only 14 years (2008-2021) in the South Amazon composite, the shortest time spam of all composites in this study. The limited length of the periods should be kept in mind, as they are shorter than the long-term dimming/brightening studies performed in regions like Europe.

We used cloud cover data from in situ measurements, clear-sky SSR time series derived with two different methods (using Synop cloud cover and using the method by Correa et al., 2022), atmospheric absorption calculated combining in situ and satellite measurements, AOD from the CAMS reanalysis, AAOD from OMI satellite observations and anthropogenic emissions from EDGAR to investigate the causes of the SSR trends in the eight composites in their period of data availability. All in-situ data went through quality control procedures to attest their validity and documentation of gridded data was carefully considered to account for potential issues. Our results showed that a strong dimming occurred in the composites located in north and northeast Brazil (Manaus, Belem, Fortaleza and

Salvador) in the period of study, while the other four composites all showed statistically insignificant
trends (positive in the South Amazon and South Brazil, and negative in the Southeast and Midwest).

A detailed analysis on the data revealed significant contributions of both clear-sky SSR and
cloud cover changes to the trends observed in the north and northeast Brazil, but with a dominance of
the effects of increasing atmospheric absorption under all-sky conditions. This is believed to be
associated with increased anthropogenic/urban emissions, which would also explain the clear-sky SSR
dimming, and the characteristics of cloud occurrence in those regions. Previous studies (e.g, Li et al.,
1995; Byrne et al., 1996) have discussed the increase in atmospheric absorption under broken cloud
conditions due to the multiple scattering by clouds and absorption by water vapour and aerosols. The
massive occurrence of partially cloudy conditions at these regions, in comparison with the other
composites analysed in this study, make this mechanism much more relevant at the North and Northeast
Brazil stations than in all of the others. The importance of changes in atmospheric shortwave absorption
to dimming and brightening have also been highlighted by Schwarz et al. (2020). Even though in the
present study we have been able to identify different factors significantly affecting SSR trends in these
regions and their magnitude, future work would be important to more precisely quantify the
contributions of each factors causing SSR changes and to project the contributions of these factors in
the future.

In Southeast and Midwest Brazil, statistically insignificant negative SSR trends in the period
indicated that no single strong forcing dominated dimming and brightening. Therefore, the small trends
were most likely the result of competing effects of negative cloud cover trends (resulting in a positive
forcing on all-sky SSR) and negative clear-sky SSR trends (resulting in a negative forcing on all-sky
SSR), where the clear-sky trends are also most likely associated with changes in aerosol absorption
(due to changes in anthropogenic/urban aerosols). In the South Amazon the signal of the strong aerosol
reduction, resulting from the reduction in biomass burning in the Amazon at the beginning of the 21st
century (Silva Junior et al., 2021), dominated the observed brightening. This AOD reduction covered a
large area in central Brazil, but did not reach most of the other composites, and areas with the strongest
negative AOD trends (stronger than -0.06 per decade) were all located in the South of the Amazon.
However, the resulting SSR trends (both all-sky and clear-sky) were not statistically significant. A
potential reason for this might be the strong seasonality of the biomass burning in the Amazon (Schwarz
et al., 2019), which means that the strong changes in AOD are affecting SSR only a few months per
year, during winter and spring months (which might not be massively relevant since the stations in the
composite are around 10-15 degrees south). Due to missing data we were not able to assess the extent
of cloud cover contribution to this result. Finally in South Brazil, competing minor effects of cloud-free
processes and cloud cover changes resulted in statistically insignificant brightening.

This study contributes to the understanding of the causes of SSR decadal trends in a world
region with still limited observational data. Further research would, however, be largely relevant,
especially in the quantification of each of the factors causing SSR changes and in the estimation of these
factors in the future.

## Data availability

The data from the IAG/USP station can be requested at
http://www.estacao.iag.usp.br/sol_dados.php (last access: 21 Feb 2024). The data from
INMET stations can be requested at https://bdmep.inmet.gov.br/ (last access: 21 Feb 2024).
The BSRN SSR data is available at the BSRN website (https://bsrn.awi.de/). The CERES
products are available at the CERES website (https://ceres.larc.nasa.gov/data/). The ERA5
reanalysis data used in this study is available under
https://cds.climate.copernicus.eu/cdsapp#!/dataset/reanalysis-era5-single-levels-monthly-
means . The CAMS AOD reanalysis data is available under
https://www.ecmwf.int/en/research/climate-reanalysis/cams-reanalysis . Data of
anthropogenic emissions estimates is available at the EDGAR website
(https://edgar.jrc.ec.europa.eu/emissions_data_and_maps , last access: 21 Feb 2024). The data
from the OMI instrument used in this study is available at
https://disc.gsfc.nasa.gov/datasets/OMAERUVd_003/summary (last access 21 Feb 2024). The
satellite cloud fraction data from CLARA, used to apply the clear-sky method used in this
study, can be found on the CM SAF website (https://www.cmsaf.eu/) and downloaded using
the Web User Interface at https://wui.cmsaf.eu/

## Author contributions

LFC designed the study, organised the data and wrote the original manuscript. DF, BC and MW revised
and edited the text. All authors contributed to the analysis and to the final paper.

## Competing interests

The authors declare that they have no conflict of interest.

## Acknowledgements

This study was funded by the Swiss National Science Foundation grant no. 200020_188601.
The authors would like to thank the Instituto Nacional de Meteorologia (INMET) and the Weather
Station of the Institute of Astronomy, Geophysics and Atmospheric Science of the University of São
Paulo for providing the meteorological observations. We express our gratitude to the teams that produce
and maintain the high quality meteorological data used in this study, from BSRN, CERES, ERA5,
CAMS, OMI, EDGAR and CLARA.

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

trends of downward shortwave radiation over Brazil. *Atmospheric Research*, *250*,
1053     105347.

**Appendix**

| Station | Composite | Coordinates | Does it include SYNOP cloud cover? | % of monthly data available |
|---|---|---|---|---|
| Manaus | Manaus region | 3.10S 60.01W | yes | 87 |
| Coari | Manaus region | 4.10S 63.14W | yes | 74 |
| Rio Urubu | Manaus region | 2.63S 59.60W | no | 75 |
| Urucará | Manaus region | 2.53S 57.75W | no | 82 |
| Belém | Belém region | 1.41S 48.43W | yes | 95 |
| Castanhal | Belém region | 1.30S 47.94W | no | 71 |
| Tucuruí | Belém region | 3.82S 49.67W | yes | 92 |
| Salinópolis | Belém region | 0.62S 47.35W | no | 69 |
| Alta Floresta | South Amazon | 10.07S 56.17W | no | 62 |
| Ariquemes | South Amazon | 9.94S 62.96W | no | 73 |
| Juína | South Amazon | 11.37S 58.77W | no | 78 |

| | | | | |
|---|---|---|---|---|
| Porto Velho | South Amazon | 8.79S 63.84W | no | 73 |
| Sorriso | South Amazon | 12.55S 55.72W | no | 85 |
| Fortaleza | Fortaleza region | 3.81S 38.53W | yes | 68 |
| Areia | Fortaleza region | 6.97S 35.71W | yes | 85 |
| Caicó | Fortaleza region | 6.46S 37.08W | yes | 68 |
| Natal | Fortaleza region | 5.83S 35.20W | yes | 65 |
| Brasília | Midwest | 15.78S 47.92W | yes | 84 |
| Goiânia | Midwest | 16.64S 49.22W | yes | 94 |
| Campo Grande | Midwest | 20.44S 54.72W | yes | 79 |
| Salvador | Salvador region | 13.00S 38.50W | yes | 89 |
| Cruz das Almas | Salvador region | 12.67S 39.08W | yes | 66 |
| Feira de Santana | Salvador region | 12.19S 38.96W | yes | 64 |
| Itirucu | Salvador region | 13.52S 40.11W | yes | 61 |
| Curitiba | South | 25.44S 49.23W | yes | 72 |
| Porto Alegre | South | 30.05S 51.17W | yes | 95 |
| Santa Maria | South | 29.72S 53.72W | yes | 89 |
| Florianópolis* | South | 27.60S 48.52W | yes | |
| Campos do Jordão | Southeast | 22.75S 45.60W | yes | 69 |
| Monte Verde | Southeast | 22.86S 46.04W | no | 84 |
| Rio de Janeiro - Marambaia | Southeast | 23.05S 43.59W | yes | 78 |
| Seropédica | Southeast | 22.75S 43.68W | no | 98 |
| São Paulo* | Southeast | 23.65S 46.62W | yes | 99 |


**Table 3: Stations used in the study, the composites they were associated with, their coordinates,**
**information whether Synop cloud cover data was available and the percentage of months with**
**available data (out of all the months in the period used for the respective composite - see table**
**1). *Stations not from the Brazilian National Institute of Meteorology. Florianópolis station from**
**BSRN; São Paulo station from the Institute for Astronomy, Geophysics and Atmospheric**
**Sciences at the University of São Paulo.**

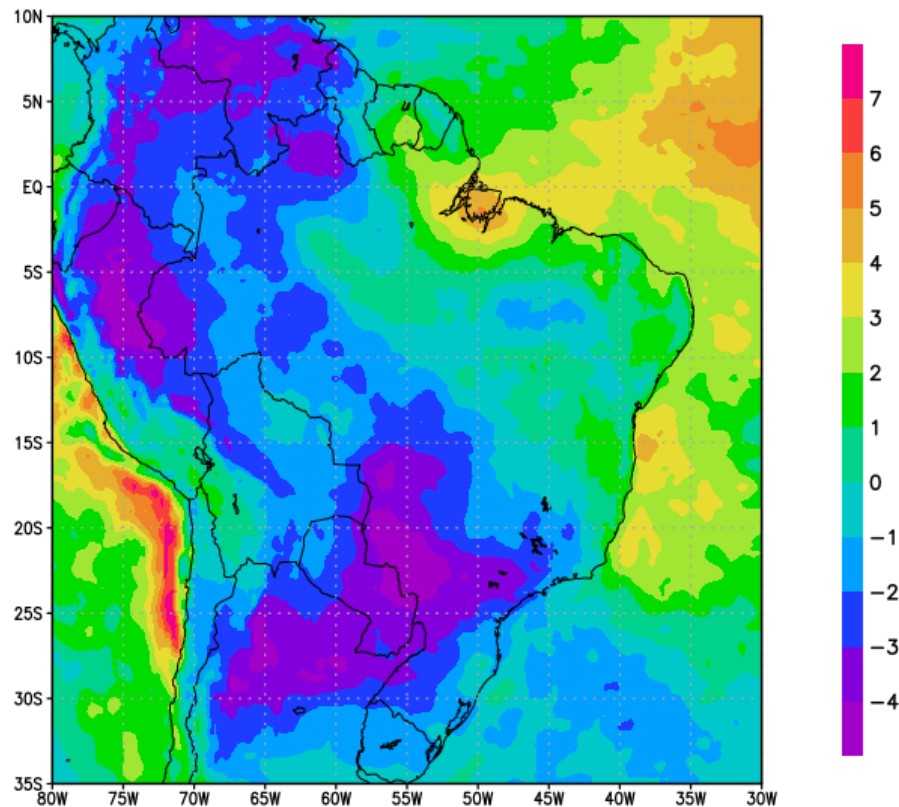


**Figure A1 - Total cloud cover trends for the 1990-2006 period (in % per decade) from ERA5.**