# Peer review of "Trends in observed surface solar radiation and their causes in Brazil in the first two decades of the 21st century."

_EGUsphere, 2024_

## Author Response (AR1)

Dear Dr. Stelios Kazadzis,

Thank you very much for the reviews on the manuscript "Trends in observed surface solar radiation and their causes in Brazil in the first two decades of the 21st century". We highly appreciate the reviewers efforts in highlighting important aspects and providing constructive feedback on the manuscript. We took most of the suggestions into account and took special care to respond to each point of concern by the reviewers. The changes are highlighted in the manuscript.

Here is a point by point response to the reviewers' comments and concerns.

Color code:
Red: reviewer's comment.
Black: authors' response.

**Reviewer #1**

This study aims at investigating the trends in solar radiation over Brazil and their causes over a period of about 20 years using data from 34 stations distributed over the country. A wealth of information from different independent sources is employed to explain the observed trends under all-sky and clear-sky conditions. The discussion of the results is based on a combination of qualitative and quantitative estimates of changes in different factors. Despite some issues that are addressed in my detailed comments below, the paper is well written and its findings are useful for understanding the recent variability of surface solar radiation over Brazil. The size of the country with different climate regimes and diversity of activities offer a good opportunity to investigate how different factors can influence solar radiation. I find this paper's contents suitable for ACP. However, it its current form I cannot recommend acceptance for publication.

**General comments**

Two decades of data cannot justify the word "Decadal" in the title and generally in the manuscript. It can be simply changed to "Trends". For some figures even less than 20 years of data are shown and discussed. The fact that trends are given per decade does not justify the term decadal in a broader sense.

We thank for the suggestion. To attend the reviewer's suggestion we removed the word "decadal" from the title, as the end of the title provides a time reference for the analysis ("the first two decades of the 21st century"). However, we still keep the word "decadal" in most instances it was already mentioned in the text. We find the word relevant as it sets a time reference. That is, we are not looking at a trend over days, months, or a few years, we are looking at trends over periods longer than a decade (14-22 years in the study), and the word "decadal" provides this information within the text.

The regional composites include, in most cases, a station installed in a big city where the irradiance levels should be expected to be lower compared to the rest of the stations. This

means that the multi-station average of irradiance is biased low by the city station. This has not been discussed in the context of effects arising from factors dominant in the city stations. It would be interesting to see the variability of each yearly value in the trend-plots by adding either standard deviation bars, or high-low ranges, or even the individual data points from each station in smaller symbols of different color.

The use of big cities as a reference to construct the composites used in this study were necessary, because most of the INMET stations with longer time series and less missing data are located in or around big cities. This information is now included in section 2.5. The only composite not centered around big cities is the South Amazon composite, and that is the composite with shortest time series and most missing data.

The potential impact of the big cities to the composites is mentioned in the end of section 3.3:

"Even though, according to figure 8, anthropogenic emissions did not increase significantly in the Amazon region, emissions still increased around the biggest cities in the region, like Manaus and Belem. This is of special relevance for this study, since seven of the eight composites are centered around cities with over a million inhabitants, where the large and usually increasing population (Lobo and Cunha, 2019) plays an important role in the atmospheric composition. The only composite that does not follow this rule is the South Amazon composite, where the biggest city is Porto Velho, which in 2020 had a population of less than 500'000 people (IBGE 2022)."

And in the end of section 4.2:

"All of this points to a relevant influence of anthropogenic factors on the SSR trends in the first two decades of the 21st century in the regions around Manaus, Belem, Fortaleza and Salvador. Remembering that these are all big cities with over a million inhabitants each, therefore this result could be biased towards big population centers."

Nevertheless, we believe that showing the effects of highly populated regions is also important, since studies in other regions of the world (i.e. China, India, Europe) found relevant contributions of urban-industrial emissions to observed SSR trends.

I am concerned about the use of the second method to identify days under clear-sky conditions. The cloud cover data are available in only two instances in the day (12 and 18 UTC) which challenges their representativeness for the entire day. I doubt that the method is trustful. The trends shown in Table 1 derived by the two methods are rather inconsistent and it seems that the second method is applicable only in 3 composites and only in 2 cases shows consistent results with the other method. Therefore I don't see the value of using this synoptic cloud cover method.

The procedure of performing cloud cover measurements at the meteorological stations at the so called "Synoptic times" (0000, 0600, 1200 and 1800) follows the World Meteorological Organization guidelines and is intended to be representative for the day. In the case of this study we are interested in daytime only, so we consider the two measurements occurring during daytime.

This is usually a reasonably approximation, as any conditions or phenomena at the mesoscale or higher scales will have a timescale larger than the 6-hour interval between

the observations. Smaller scale phenomena are a problem in this context, since the process of cloud formation tends to depend on smaller time-scales. This could be a problem for the regions that we highlight being more dependent on local convection for cloud formation, in the north and northeast Brazil. But as a consequence of this stronger dependence (relative to the regions further away from the equator) on local convection, these locations will have fewer occurrences of clear-sky conditions. This can be seen in figure 9. This happens because very often the formation of broken clouds will be continuous throughout the day.

So, in summary, when large scale phenomena dominate cloud formation, the two diurnal observations should be representative for the daily conditions. On the other hand, when local phenomena dominate cloud formation, very often there will be no occurrence of completely clear-sky days in the regions that we analyzed.

In addition, using cloud cover observations to identify clear-sky conditions is the most simple method often used to identify clear-sky conditions. With this in mind, we still see a value in the use of this method, especially considering that it is used in addition to other clear-sky methods. But, even though we here presented the reasoning for its consistency, this method, as any other clear-sky method, is not perfect. So we believe that some of the issues indicated by the reviewer might have played a role for the synop clear-sky time series in the southeast Brazil. However, we should highlight that, even with the remarkable differences in the inter annual variability, the synop clear-sky time series of this composite agrees with the other clear-sky method in terms of direction and significance of the trend.

Specific comments

18: The term "clear-sky processes" is used repeatedly in the paper without discussing somewhere which these processes are.

Explanation of the term was included in the abstract and in the body of the text:

"(attenuation of solar radiation under cloudless conditions)" in the abstract, and

""Clear-sky processes" in this context refers to the interaction between solar radiation and the components of the atmosphere without the presence of clouds." In section 3.1.

37-57: For the studies cited in the Introduction, it would be helpful for the reader to include the period each study has investigated.

The information was included whenever possible.

70: It would be helpful for the reader to point to Table A1 also in this place.

Done.

84: It would be useful information to state what percentage of the hourly, monthly or yearly data were missing, not necessarily for each station but, for example, for each region, or at least on average for all stations.

We included in table 3 one column with the information of the percentage of months with available data, out of all the months in the period used in the analysis of the respective composite. For example, the South Brazil composite covered the 2001-2022 period, with a

total of 264 months. The Santa Maria station had valid data in 234 of these months, resulting in 88.6% availability - which is rounded to the nearest integer in the table.

119-121: Why such a small criterion (only 2 days per month) was used in constructing monthly mean data of AAOD? Two days per month can hardly be considered representative for the monthly value of AAOD. How much the dataset would be affected if a more representative criterion (e.g. 5-10 days in a month) would have been adopted?

We used a small criterion because less than 30% of the days had valid values of AAOD. This happened because AAOD is only retrieved without the presence of clouds. Thus we used a similar criterion as we used for clear-sky identification. If we apply a criterion of 5-10 days in a month we will not be able to derive a time series out of it, due to missing data.

133: Although the effect would be rather small, has the conversion from m2 to grid taken into account the variation of the area of the grids due to earth's curvature?

We thank the reviewer for drawing the attention to this. But yes, this was taken into account.

251: Please state what are the confidence levels that are used in the analysis (e.g. 95%).

We Apologize for the missing information. Now this information (95%) was included throughout the manuscript.

264: The linear trend lines could also be shown on these plots, as well as in Figures 3 and 4.

We updated the figures including the linear trends, as requested.

269-279: Please make this caption clearer. The text is too long and difficult to follow. Some information could be moved or already are in the main text (e.g., the clear-sky methods, the reference to Table A1. What are the numbers in square brackets under Synop Cloud cover?

We now include the information about the square braces under Synop Cloud cover in the caption. However, we opted to keep the long and detailed caption as it carries all relevant information to understand the table.

296: Make clear that this statement refers to SSR "based on synop cloud cover" only.

The sentence was changed accordingly.

296-302: According to Table A1, in the Southeast region not all stations provided synoptic cloud data. Why this region is treated differently from the Belem and Manaus regions and the respective synop-derived clear-sky time series is included in the plots? Moreover, for this region the agreement of the two methods worse both in terms of trends and the variability of the time series.

The reason for the different treatment of the composites is now explained in the text: "Synop clear-sky time series were derived when at least three stations in the composite had clear-sky data (see availability in table 3). Manaus, Belem and South Amazon composites did not fulfill this requirement."

Regarding the agreement in the SE Brazil composite, this is discussed in the text as follows: "For the composites where clear-sky data is available from both methods used, in two of

them (South and Midwest Brazil) both methods indicate very similar inter annual variability and trends, while in the other (Southeast Brazil) the two methods do not show strong agreement in the inter annual variability, but agreed in the direction of the trend. Therefore, the results of the clear-sky SSR trends are supported by both clear-sky methods."

309-310: In fact, in only 2 stations the clear-sky trends with the two methods are similar. This is another indication that the synop method is questionable and does not provide consistent results in 6 of the 8 regions.

This aspect was addressed in the response to the previous comment.

324: How the cloud cover data from different stations were combined to make the regional averages shown in Figure 4? Did you use simply the arithmetic mean? Generally I would be reluctant to averaging cloud cover data as yearly means, especially if these are used to compare trends in irradiance. Could effects on irradiance are non-linear and the variability of their averages (even as daily means) should not be directly comparable.

Yes, we used arithmetic means following the procedure described in section 2.1.

We agree that the cloud effects on irradiance are non-linear. Therefore, the precise estimation of the value of cloud radiative forcing would require more detailed information. But in general terms, one can safely state that an increase in cloud cover has a negative forcing on SSR (and a decrease in cloud cover has a positive forcing on SSR). In most locations of the world one can identify a strong correlation between SSR and cloud cover even on the annual time scales. For this reason, we perform the comparison between cloud cover and SSR as part of our analysis. This was also done by previous studies referenced in our manuscript (e.g. Stjern et al., 2009; Norris and Wild, 2007).

328: As cloud observations show large spatial variability, I would suggest to show in Figure 4 also the data from the individual stations (with smaller symbols of different color). You could also consider overlaying the data of Figure 2 to allow a direct comparison of the variability and the trends, keeping in mind the previous comment.

We updated the figures according to the request. Now figures include error bars which indicate the range of the values of the individual stations in that composite for the respective year, and dashed lines which indicate the trends.

331: Comparing the trends of Figures 2 and 4 is based on the assumption that the cloud cover observations in each region are representative for all stations in the region, even for those without synop observations. Please discuss this briefly in the text. Note that this assumption has not been applied in the data of Figure 3.

The following paragraph was included in the new version of the manuscript to address this issue:

"Manaus, Belem and Southeast Brazil composites do not have synop cloud cover data for all stations (2 out of 4 available for Manaus and Belem, and 3 out of 5 for Southeast Brazil), therefore a comparison between all-sky SSR trends and Synop cloud cover at these composites is based on the assumption that the cloud cover observations for the composites are representative for all stations. This is a reasonable assumption given the geographical proximity between the stations within these three composites and the lack of any climatic or geographical feature that can strongly affect cloudiness at individual stations (e.g. high topography). In these composites, all stations are

located in areas with the same precipitation regimes as classified by Ferreira and Reboita (2022), also corroborating with the assumption of good representativeness."

340-341: Please elaborate a little more on the statement about the relevance of the CCRE on the derived SSR trends.

The CCRE is better described in the following part of the text included in the new version of the manuscript:

"That is, the cloud cover trends are too small and, and as a consequence, the contribution of changes in cloud cover to the SSR trends is expected to be minor. This contribution is estimated objectively by the CCRE (see table 1), which shows, in most cases, low values (in comparison to the all-sky SSR trends), suggesting only a minor contribution from cloudiness to the SSR trends."

386: The term (0-1) in the y-axes titles is misleading since the plots show anomalies. It is better to remove it. Drawing the trend lines would help also in this figure.

The figures were changed accordingly in the new version of the manuscript

412: Since the OMI AAOD is used, I wonder why the AOD from OMI was not used also in Figure 5a instead of the CAMS reanalysis, in order to maintain consistency between the two data products.

In this study we tried to chose to prioritize observational data (in-situ or satellite based) when it came to irradiance and reanalysis data when it came to anything else (except for the station observed variables, which were, of course, observed). Thus, we originally chose to use CAMS reanalysis AOD. We decided to analyze OMI AAOD because this variable was not available in the reanalysis.

The reason we chose to prioritize reanalysis data for supporting variables (AOD and water vapor) was to avoid problems with missing data. This is a recurring issue of AAOD from OMI, which has missing values when the scene was too cloudy to perform the aerosol retrieval.

414: Figure 7 shows the AAOD in relative units while Figure 5a in absolute units, so direct comparison is very difficult. I suggest using absolute values in the trends.

We thank the reviewer's suggestion. We updated figure 7 in the new version of the manuscript. Now it shows two maps, one for absolute and one for relative trends.

**Technical**

The verb "fit" is used repeatedly (manly in the discussion sections) to indicate consistency or compliance between results or findings. It is better to be replaced with expressions like "comply with" or "consistent to".

Changes were made accordingly throughout the text.

25: Replace "object" with "subject"

Done.

82: Replace "went" to "were"

Done.

193: Replace "occurred" with "are"

Done.

210: "both locations" which are those? Belem and Manaus?

Yes. This was clarified in the text.

265: I suggest rephrasing as follows: "In each composite, anomalies are in reference to the mean of the entire period (shown in table 1)."

Done.

281: Remove "composites" in parentheses, as already exists before the parentheses.

Done.

390: Replace "FABas" with "Fabs"

Done.

402-403: "magnification of the effects" is a bit misleading; better replace it with "magnification of trends"

Done.

426: Replace "verified" with "investigated" or "studied"

Done.

474: Replace "until where the" with "over which a" and move this explanation in line 470 where the term "decorrelation length" is first mentioned.

Done.

475: Delete duplicate "time"

Done.

494: Replace :optimize" with "increase"

Done.

539: Replace "visible" with "active" or "occurring"

Done.

551: Replace "is especially remarkable" with "occurs primarily"

Done.

568: Replace "as the" with "at the"

Done.

**Reviewer #2**

Review of the paper "Decadal trends in observed surface solar radiation and their causes in Brazil in the first two decades of the 21st century" by Correa et al. The authors present a study where they have investigated the long-term variability of surface solar radiation in Brazil in the first two decades of the 21st century. The study deserves to be published

considering the novelty of the treated region even if it analyses a short period and it uses a small dataset.

Below the authors can find a list of suggestions that must be addressed before publication especially related to the adopted methodology.

**Major comments:**

- Lines 29-33: the authors should cite an higher number of papers demonstrating to know the literature and supporting in this way the subsequent sentence " However, many regions of the world…".

We included a list of relevant studies to some of the regions mentioned in the original text. The text now reads as:

"Several studies have followed presenting the trends and discussing their causes and potential consequences in several parts of the world (Wild, 2009), like Europe (e.g. Manara et al., 2016; Norris and Wild, 2007; Power, 2003), North America (e.g. Liepert 2002), China (e.g. Feng and Wang, 2019; Wang et al., 2015), Japan (e.g. Kudo et al., 2012) and New Zealand (Liley, 2009). Global dimming (negative trends in SSR) and brightening (positive trends in SSR) have been associated, in most of the cases, with changes in cloud cover (e.g. Stjern et al., 2008; Augustine and Capotondi, 2022) and changes in aerosol loadings (e.g. Wild et al., 2021, Kambezidis et al., 2012), with the dominant aspect depending on regional atmospheric and emission features."

- Line 42: "machine learning methods…" another study on this topic can be cited (https://doi.org/10.5194/essd-15-4519-2023). It applies machine learning methods using both ground-based observations and reanalysis dataset.

We thank the reviewer for the paper suggestion. It is indeed a very interesting study, and it was now included in the references of the present manuscript.

- Section 2: how have you checked the homogeneity of the SSR series?

The stations from BSRN and from IAG/USP had full metadata available and no discontinuity reported. INMET stations are provided with information whether the station is still operating or if it has faced any mechanical failure, and we only used stations with no problems reported. On top of that, we applied the penalized maximal F test by Wang (2008) to verify the occurrence of any non reported inhomogeneity and no problems were found in the stations and periods we chose for this study. All of this information is now included in the first two paragraphs of section 2.1.

- Lines 70-84: it is necessary to add more information about the steps to move from the hourly values to the annual values in order to prove the robustness of the applied method. If any you can also add a reference of another paper where this method has been applied.

We included some additional information in the first paragraph of section 2.1. In the last sentence we point to references that used similar methodologies to cover from hour to daily, monthly and annual values: "The averaging procedure from daily to monthly, and from monthly to annual values reproduces similar methodologies used in previous studies (e.g. Stjern et al., 2008; Manara et al., 2016)."

- Figure 1: it would be interesting to see the orography of the region instead of the green colour to notice if the stations are located in plane or mountain areas. This

could also help to understand how much the selected station are representative of the composite they belong.

Figure 1 was updated and now includes topography information.

- Line 251: considering the not so long period covered by the SSR data, why did you not calculate the SSR trends with Sen-Theil? Have you verified that the trends calculated with the two methods are similar?

We made the decision of the method for trend calculation based on the distribution of the residuals. The residuals of the linear regression of the SSR trends were always normally distributed, therefore we used the LLS regression. That was not the case for all time series of cloud cover, therefore we used the Sen's slope (also called Sen-Theil). As for curiosity when we calculated the trends for SSR we also checked the trends using the Sen's slope and the results were almost identical.

- Line 258: have you considered the series as absolute series or have you calculated the anomaly series? You should specify this point.

We included a sentence in the beginning of section 2.6 that better explains this. It reads as follows: "The trend analysis was based on annual anomalies of SSR. To calculate the annual anomalies, the absolute SSR annual values were subtracted from the average SSR value for the whole period of data availability for the respective composite (see table 1). This did not affect the trends, but facilitated the visualization and comparison between time series, since the anomalies are centered around a common value (zero)."

- Caption figure 2: in the text there is not explained that the series are treated as anomalies. It should be specified. Moreover, which is the reference period? How have you treated the missing values?

We thank the reviewer for the attention on the clear explanation. In the response to the previous comment we included information about the series being anomalies and about the reference period. Information on how missing data was treated can be found in section 2.1, in the part that reads as:

"The hourly values were further converted into daily means by simply averaging the 24 hourly values in a day. If one hourly value was missing (due to either lack of data or removal during quality test) the one hourly value was filled linearly using the previous and next hours and the daily value was the average of 24 hourly values (23 observed and 1 filled linearly). If more than one hourly value was missing, the daily value was not calculated. Daily values were further converted into monthly values by simply averaging the daily means within the same month. Monthly values were only calculated when at least 70% of the days in a month were available. Further conversion from monthly to annual values again occurred by simply averaging the 12 months. If one, two or three monthly values went missing, the long term mean (mean for the whole period with available data) for that month would be used instead, and the annual mean was still calculated. If more than three monthly values were missing, then the annual value was not calculated."

- Line 290: even if the differences are evident between different regions it should be considered that they cover different periods and all of them cover less than 30 years so more caution is necessary when the regions are compared.

We included one statement immediately after table 1 and in the conclusion section to highlight this point. But the period analyzed should be long enough to start identifying the relevant aspects for SSR trends on timescales of a decade and beyond.

Minor comments:

- Line 70: "is collected and controlled by the Instituto…" add "by"

Done.

- Line 164: there is a reference for this method?

We do not know a specific reference for this, we just applied simple concepts of radiation in the atmosphere. By this correction we are simply trying to answer the question "How much would be the measured clear-sky SSR on day X (where X ~= 15) if the incoming irradiance at TOA of day X was equal to the incoming irradiance at TOA on the 15th day of the month?". This leads to the same result as if we calculated the atmospheric transmittance of day X and multiplied it by the incoming irradiance at TOA of the 15th day of the month.

- Line 181: change "absorved" with "absorbed"

Done.

- Table 1: columns 7-8 there are too many digits

We removed one digit in both columns.

**Reviewer #3**

**Summary and general comments**

The manuscript "Decadal trends in observed surface solar radiation and their causes in Brazil in the first two decades of the 21st century" explores a set of data to describe and explain trends in surface solar radiation (SSR) for different regions of Brazil, including few major cities, during the last 15 to 20 years. Compared to other regions of the world, indeed South America lack more studies on this topic. The short time series used limits the needed long-term analysis for the region, but the article brings to the discussion important aspects that may are under controls of the contemporary trends in SSR. Trends in aerosols particles loading and properties along cloud cover are the main factors used to explain trends in SSR. Surprisingly, the authors did no show any timeseries of AOD as they did for cloud cover. AOD trend is important, but it does not show important steps in AOD evolution over Brazil, which I consider to be important to the interannual variability analysis.

We opted to show the AOD trends in maps. We did this because this could facilitate the visualization of the spatial distribution of the trends. This could not be done for SSR or cloud trends as both variables come from point (in-situ) measurements (while for AOD we used gridded data). Therefore, for these variables we only show the time series.

We still believe that the trend maps allow a valuable overall spatial visualization of the trends, however, we understand the concern of the reviewer. To help the interpretation of the AOD results, in the new version of the manuscript we highlight the areas with

statistically significant (with 95% confidence level) trends. In this way we can better analyze the AOD trend map and interpret it in the context of the SSR trends.

The authors use anthropogenic emissions in way that seems to exclude biomass burning emissions or to classify it in a different emission category, smoke emission in Brazil is mostly driven by human activity. At many parts of the manuscript, urban and industrial emissions would fit better than anthropogenic emissions.

We thank the reviewer for the suggestion. Changes were made throughout the manuscript starting with the points raised in the specific comments.

The article is well organized, but the author may resort to more table to summarize their dataset and details, since there are many sources of data with a variety of features. The role of each product must be clear from the methods topic.

We thank the reviewer for the suggestion. Instead of another table we included at the end of sections 2.1 and 2.2 (the two sections which are describing the data used) a paragraph which summarizes the role of each dataset in 2-3 sentences. In section 2.1 it now reads as:

"The SSR data described in this section is used to estimate the SSR trends presented in table 1, and to calculate the fraction atmospheric column absorption (see section 2.4), which also has the trends presented in table 1. The cloud cover data described in this section was used to estimate cloud cover trends presented in table 1 and to apply one of the two methods for clear-sky identification used in this study (see section 2.3)."

And in section 2.2 it now reads as:

"AOD, AAOD, water vapor and anthropogenic emissions data described in this section were used to identify spatial distribution of the trends for these variables. TOA incoming and outgoing irradiance data described in this section was used to estimate fractional atmospheric column absorption (see section 2.4). For all gridded data described in this section, the stations were sampled by taking the gridpoint containing the station coordinates."

Before recommendation to be published, I would suggest the authors to think through their approach more carefully and analyze in more detail some aspects that I highlight in the specific comments below.

**Specific comments**

**Line18:** Clarify what do you mean by "clear-sky processes"

The sentence now reads as "In the Southeast and Midwest regions of Brazil near-zero trends resulted from competing effects of clear-sky processes (attenuation of solar radiation under cloudless conditions) and strong negative trends in cloud cover."

Line18: South of Amazon is still part of the North region; why was it treat separately? How does it compare with other parts of Amazon within north region?

This choice and the all the others are discussed in section 2.5. Specifically about the South of the Amazon, the composite was chosen to cover the region with the strongest influence of biomass burning from the Amazon, and this is mentioned in the text: "The

composite (5), South Amazon, was chosen to cover the region under the strongest influence of biomass burning aerosols from the Amazon (Artaxo et al., 2006)."

Line26: I guess evidence that SSR is not constant are older than that. You may adjust the sentence to emphasize the role of the cited studies pioneering the studies that try to understand the trends in SSR over time, not that evidence that it is not constant.

To attend the suggestion, we adapted the sentence, which now reads as follows: "Decadal trends in surface solar radiation (SSR) have been the subject of study since pioneering studies in late 1980s and early 1990s dedicated efforts to try to understand the long-term variation of SSR (Ohmura and Lang, 1989; Russak, 1990; Dutton et al., 1991; Stanhill and Moreshet, 1992)."

Line32: "Wild et al. 202" Check the year.

Thanks a lot for the heads up. The year is 2021 and it is already fixed in the text.

Line54: It would be interesting to mention what these studies say about dimming and brightening. These studies pointed out to which direction, or that was not the case?

Due to the different methods, locations and periods of the previous studies, it is hard to concisely summarize their results. We provided a general summary in the new version of the manuscript, and it reads as follows: "The studies referenced here apply different methods, to different regions, in different periods, so it is hard to directly compare them. But, in general terms, studies based on sunshine duration tend to indicate a brightening in Brazil after 1980s, while studies using machine learning techniques and regional observational studies show a spatial heterogeneity of the SSR trends in Brazil in the last few decades."

Line59-61: To which point this limited time series would tackle the existent gap. Be more precise.

To our knowledge, this is the first large scale assessment of SSR trends in Brazil using direct measurements of global irradiance covering all regions of the country. This study and future similar studies will facilitate summarizing the long-term SSR trends in the regions from the spatial and methodological perspectives (something which, as pointed out in the previous comment, is hard when different methodologies and spatial focus are given). Also, the use of measured global irradiances has the added value that we do not rely on indirect methods to estimate SSR long-term trends. We included one sentence in the text which highlights the novelty of this work: "The direct assessment of SSR long-term variability (using observed SSR) over such a large area in South America represents a novel contribution from this work."

Line70: "controlled by the instituto" instead of "controlled the instituto"

Done.

Line71: Instituto Nacional de Meteorologia instead of "Instituto Brasileiro de Meteorologia"

Done.

Line113-114: How does CAMS AOD product performs over South America? Any literature review on this?

One reference and short description is included in the new version of the manuscript: "Gueymard and Yang (2020) validated CAMS data using AERONET stations from around the

world, including South America and found that the reanalysis performs well in comparison to in-situ aerosol observations, therefore being well suited for regional and global studies."

Line119-121: Here, make clear here that this is just for AAOD from OMI.

Done.

Line123 "long term mean" it is important to clarify the time period that your long-term mean refers to, not just here but above also, when you describe the same procedure for SSR.

This information is now included (mean for the whole period with data availability).

Line125-128: Need to clarify the role of CERES data in the study.

In the new version of the manuscript this information is stated: "The data from CERES was used to estimate fractional atmospheric column absorption (see section 2.4).".

Line129 -137: All this need to have their use justified. What's the point of use Edgar if it does not include biomass burning emission, inform the reader the reason to include EDGAR, since it does not consider biomass burning emission.

The following sentence was included to clarify the reasoning for the use of EDGAR data in the analysis: "This dataset was used, even though it does not include biomass burning, because it provides information about aerosol emissions from all other sources, which are also relevant, such as urban and industrial emissions."

Line143-149: Would this threshold remain the same as time goes by? High aerosol loading events you said that would not be a problem for long term trends, but within the decades analyzed there was a strong shift in the aerosol loading in the atmosphere, mainly in the center-west of Brazil and Amazonia.

Yes, the threshold remains the same as the time goes by. But as transmittances for the whole period are used to define the transmittance thresholds, the high aerosol loading events are taken into account and the method adjusts to them. This happens because the more high aerosol loading events we have, the more days with lower transmittance will occur, affecting the distribution of daily transmittances, but not affecting the distribution of satellite cloud cover (read the method reference for more detailed information). As a result, recurring events of high aerosol loading leading to low atmospheric transmittance are still well identified. The weakness of the method is the outliers: extreme events which occurred one or a few times only. These days would be identified as cloudy and flagged out of the clear-sky time series. Therefore, short-term variability (from days up to months or even a few years) could be misrepresented, but the long-term variability (beyond a decade) is well captured.

Line155-157:  How does this fit with the first method threshold?

The methods are independent. But both use a "trade-off": instead of flagging only looking for only 100% clear days (0% cloud cover), both look for mostly clear days (in the case of the synop cloud cover method, >80% clear). This is done to allow more data and facilitate the construction of a time series, as the 100% clear days are much more scarce. In the end, the occurrence of days flagged as clear-sky is station dependent, but tends to vary between ~5-35% of all the days in the time series in both clear-sky methods.

Line 171-174: You are working with products with different resolutions, at certain point will be important to describe how did you manage to sample this product around each station. Did you take the pixels that contain the coordinates of the station, or you did an average over a specific area? It would be helpful if you organize these products in a table with their description.

We clarified the sampling around each station in the text at the lines indicated by adding the sentence "For the gridded data the pixel containing the station coordinates was used.". This information was also included at the end of section 2.2.

Line 175: I guess it will make it easier by separate the first term as absorption produced by the system (atms + surface) and the second term just the absorption produced by the Surface. A question, did you describe the source for TOA SW?

We thank the reviewer for pointing out the lack of description of the source for TOA SW. We used a CERES product, and the new version of the manuscript points to the correct product.

Regarding the terms in equation 1, the first term in fact refers to the reflection at TOA, i.e. all the solar radiation lost back to space. It is the result of the outgoing SW at TOA divided by the incoming SW at TOA. The second term refers to absorption at the surface, as it is the result incoming SW at the surface divided by the incoming SW at TOA, all this multiplied by the surface absorption (1-albedo).

Line187: In your figure one you could take a climate basis classification to support objectively this statement. My point, you must support in clearer way this coverage of different climate characteristics.

Figure 1 now also includes topography information. The entire section 2.5 is already dedicated to describe all the different climate characteristics, by pointing to references and discussing the important atmospheric mechanisms in each region.

Line195-197: It would be helpful to include Reboita et al. climates domain to contextualize the positions of your sites.

We thank for the suggestion, however we do not have access to the data used in the study by Ferreira and Reboita (2022), where they applied a clustering analysis to separate the regions. Therefore, the most we could do would be drawing approximate lines, without a guarantee of accuracy. For this reason we did not include visual indications of the regions, but we reference the paper, which has the maps with the regions well indicated. We also discuss in the text where the composite are located.

Line205: You are mentioning that precipitation is tied to local and mesoscale, but soon you bring ITCZ as the most important large-scale elements to explain precipitation seasonality in the region, which is true. Suggest you adjust the sentence, so local, regional, and large-scale role can be equally acknowledged.

We updated the part of the text to include this information. Now it reads as follows: "Precipitation and cloudiness in both regions is strongly tied to local to mesoscale phenomena, like local convection, sea breeze circulation and squall lines. At the large scale, the Intertropical Convergence Zone (ITCZ) also has a significant influence on the precipitation in the regions, playing a major role for the seasonality of precipitation (Fisch et al., 1998)."

Line 212-213: It is true that Manaus and Belem are not strongly influenced by the most important biomass burning region in the Amazon, which is the southern portion of the rainforest ecosystem, but Manaus and Belem are also affected by smoke, mainly from the biomass burning season in the northeast portion of Amazon, and AOD are really significant at periods (See this: https://amt.copernicus.org/articles/12/921/2019/)

We apologize for the misunderstanding. In fact we did not intend to say that Manaus and Belem have no influence of biomass burning aerosols, the intended statement is to say that both Manaus and Belem have LOWER influence of biomass burning aerosols than the southern Amazon. This is now more clearly stated in the manuscript: "The occurrence of the South American Low Level Jet (Vera et al., 2006), important for moisture and aerosol transport from the Amazon to Southeastern Brazil, leaves the locations of Belem and Manaus with lower influence of biomass burning aerosols than the southern fraction of the Amazon."

Line221: Try to display these regions in figure 1, it will make it easier for the reader.

Three comments ago (line 195-197) we addressed this topic.

Line235: It depends on the season, during summer local convection associated with sea breeze plays a major role in cloud diurnal cycle.

The sentence was updated to include this information and now reads as follows: "Like the Middle West and South Amazon composites, cloud formation in this region is mostly associated with large scale phenomena, with significant influence from local convection and sea breeze being limited mostly to summer months (Reboita et al., 2010; Ferreira and Reboita, 2022)."

Line238: smoke aerosols are also anthropogenic, at least those from biomass burning in most of Brazil, so I'll recommend you replace anthropogenic here to Urban-Industrial emissions.

We thank the reviewer for the suggestion. The term was corrected in the text.

Line241; Frontal system instead of "fronts" and extra-tropical cyclones instead of "subtropical cyclones"

The text was changes accordingly.

Line243: Again, replace anthropogenic to urban-industrial emissions.

Also done.

Line261: "Figure 2 shows the all-sky SSR anomalies time series" instead of "Figure 2 shows the all-sky SSR time series."

The sentence was corrected as suggested.

Line264 (Figure 2) Why not include the standard deviations for each case. This would justify one plot for each site, otherwise I would use just one figure to plot all the sites along with different colors.

We included error bars that indicate the range of values for the individual stations that were used to calculate the composite values in this and all other time series plots.

Line318-319: Clear sky cases correspond to which fraction of all cases? Clear sky cases are expected to be associated with particular meteorological scenarios; how can these aspects affect your conclusion here?

The clear-sky cases tend to correspond to between 5-35% (station dependent) of all cases. The particular meteorological scenarios would not affect the conclusions, because we are simply analyzing SSR trends under clear-sky conditions, so the important aspect is whether there are clouds or not.

Line321: At this stage, without any analysis on aerosols, clouds etc, this sentence sounds strange.

The sentence was rewritten to avoid confusion and now reads as follows: "Further analysis is thus needed to better understand the reasons for the clear-sky and all-sky decadal SSR trends observed in Brazil."

Line342-34: "…major cloud contribution…" you have to keep in mind that it is only about cloud cover, your clouds dataset does not allow you to infer change in other aspect of clouds. (High, low, middle clouds) Have a look on this article for Sao Paulo, https://rmets.onlinelibrary.wiley.com/doi/abs/10.1002/joc.6203

We thank the reviewer for highlighting this important aspect regarding cloud effects. In the referred sentence we included the term "Cloud cover contribution" to clarify we refer to contributions originating from changes in cloud cover. We are aware of the potential of changes in other aspects of clouds, you mentioned changes in cloud types but one could also highlight cloud optical depth, for example. However, in the case of the cloud cover trends with near-zero values in the South Brazil region, any changes in other aspect of clouds did not affect significantly their occurrence. Therefore, the strongest contribution of clouds to the energy balance, which is their occurrence or not, can be already ruled out.

Line351: Why CWV is included in the analysis, I'm not sure that this was clarified previously, but it is important to explain the purpose of including CWV in this analysis.

We included two sentences iat the beginning of section 3.2 which now justify the analysis of clouds, AOD and water vapor. The sentences read as: "Clouds, aerosols and water vapour all can attenuate solar radiation, therefore, their variability is analyzed in more details in this section. The order in which they are mentioned follow the order of relevance in the discussion of solar radiation attenuation in the atmosphere, with clouds being the most important aspect and water vapour the least important aspect."

Line353 (Figure 5) Colorbar dimension is missing (all figure with colorbar dimension is missing not just this). Also, it would be important to include in these maps an indication of area were the trends are statistically significant.

We included shaded areas indicating statistical significance (at the 95% confidence level) in all trend maps. We also included the unit information in the colorbars of all trend maps.

Line 360: Do you mean southern hemisphere winter? Not summer.

Yes. The information was corrected in the new version of the manuscript.

Line363: Surprisingly the impact of the reduction of smoke from south amazon on downwind region does not appear.

We thank the reviewer for spotting the error. This was changed accordingly.

Line395-397: Negative trend for clear sky seems consistent with smoke loading trend, but which would be the explanation for positive trend under all sky conditions? It would be

interesting to see the seasonal distribution(frequency) of you days for both clear-sky and all-sky conditions.

The referred trend was statistically insignificant. Now this information is written in the text. Due to the lack of statistical significance and the lack of cloud information in the South Amazon, we did not try to further investigate the reasoning for that.

Line 405: cloudiest instead of cloudier

Done.

Line421- 424: Does this result find echoes in the literature? How about OMI uncertainty in region with lower aerosol loading? How about the significance of this trends? Smoke from biomass burning is an important source of regional absorption. I would expect a reduction in regional smoke to produce a reduction in regional AAOD.

We unfortunately did not find any related literature, unfortunately the region is limited in that sense. The OMI performance is better discussed in two responses below, but in summary the documentation does not point to issues associated with regions of lower aerosol loading.

Regarding a reduction in regional AAOD, this is in fact what we see in the south of the Amazon on figure 7.

Line446-449: Let's say that this may explain AAOD increase in the southeast of Brazil and along the region close to the coast, but how about the central and north of Amazonia, where is found the larges positive trend in AAOD (see figure 7). That's why needed to focus on areas with trends statistically significant. Again, biomass burning is also anthropogenic emissions, and it has important fraction of black carbon, south of Amazonia lower smoke would contribute to less black carbon in the atmosphere.

Thanks for the bringing this up. In the new version of the manuscript we updated figure 7, to show the trends in absolute and relative values, with the areas with statistical significance (at the 95% level) shaded. We also recalculated the trends using the Sen's slope, since we realized we haven't previously verified if the residuals were normally distributed. As a result, the value of the trends changed slightly, but the overall pattern persists. We still see the increase in AAOD in the northern of the Amazon, and now we can see that it is statistically significant. We do not have an answer for this. An educated guess would be that this is associated with the emissions in the neighboring countries, like Venezuela, Colombia Suriname and the Guyanas. The Andes in the west act as a barrier, and as a consequence the westerly flow that passes over these countries cannot easily continue to the Pacific Ocean, and as a consequence aerosols also would be "trapped" in the region. But we do not have any evidence for this, this discussion is purely speculative.

In the rest of Brazil we see many areas with statistically significant increase in AAOD, which complies with the discussion already present in the previous version of the manuscript.

Line 464-465: It also would be important to bring some notion about the performance of satellite and models products that were use.  Since they would indicate if there were known bias that can help to explain some aspects of your results.

A paragraph was included to briefly discuss this. In summary, no major issues that could significantly affect our analysis were found in the documentation of the products used. The paragraph reads as follows:

"The performance of the gridded products used used in this study are discussed in their respective documentations, referenced in section 2. Spectral surface albedo is reported as a main source of uncertainty in the satellite based products, especially OMI AAOD, however, this tends to be a major problem over the ocean. Sub-grid cloud contamination tends to also represent a problem for retrieval of satellite based products. But this is reported to lead to over/under estimation of the average AAOD, but to not affect the representation of its long-term variability. No issues with long-term variability of the reanalysis products were reported."

Line 495-497:  This was already said few sentences above.

The repeated sentence was removed.

Line502: How about the cities mentioned above, are they not influenced by mesoscale and synoptic scales?

In the sentence we refer to relative influence, that is, the cities mentioned have a lower dependence on the mesoscale and large scales than the other cities. This is now highlighted in the text: "his is not observed at all the other locations, which have a higher dependence on mesoscale and synoptic scale phenomena for cloud formation than the previously mentioned locations. ".

Line 517-529: If you could think in a diagram (with good visual perspective) that summarize this it would good.

We Thank the reviewer for the very nice idea. We could not think of an elaborated diagram, however we just reorganized the information of this paragraph in a table (table 2 in the new version of the manuscript), and that should facilitate the assimilation of the three groups and their common characteristics.

Line 574 (Figure 9)- there is a missing plot (column 1, row 2)

There was not enough data available to derive a distribution for the South Amazon composite, therefore the plot was not included. This information is included in the caption of the figure. Figure 9 is plotting the same data from figure 4, which also does not include the South Amazon plot due to missing data.

Line 655-657: But you also have to take in to account the fact that cloud cover trend was not evaluate for this region.

The reviewer is right in pointing out that cloud cover trend was not evaluated for the region and could be relevant. However, in the results presented we see statistically insignificant positive clear-sky SSR trends even though we see strong negative AOD trends. That already shows an important result which is independent of clouds.

Line 700-701: This discussion is generic about this; the author's need to search for more evidence in the literature to support this. Actually, I found that the author barely explores regional literature on their discussion, I mean studies that try to analyze trends in aerosols in South America.

Unfortunately the literature is limited in the region. This is one of the motivations on why we chose this region for this study: we lack studies in the region. Regarding studies on aerosols in South America, we find a significant amount of studies for the city of Sao Paulo involving especially air pollution and aerosol properties, and other studies that focus on the Amazon, especially on aerosol properties. But we could not find studies that analyze long term aerosol variability covering a large area of South America.

In what concerns the discussion of the results of north and northeastern Brazil, we have a detailed discussion on the physical processes and the interpretation of the results in section 4.2.

Line 711-712: This is hard to say, there are many open aspects, cloud cover for south amazon was not evaluated, which can play an important role on these results. So, I would recommend the author to be more cautious here.

We changed the part indicated following your suggestions and now it reads as follows: "But the resulting SSR trend was not statistically significant. A potential reason for this might be the strong seasonality of the biomass burning in the Amazon (Schwarz et al., 2019), which means that the strong changes in AOD are affecting SSR only a few months per year. Due to missing data we were not able to assess the extent of cloud cover contribution for this result."

Line714-716: That's true, however, the limited time frame of the time series is still a challenge for SSR trend evaluation as has been done for north hemisphere regions.

It is hard to compare such an observational study in South America with studies in the northern hemisphere, because observational networks have been established much earlier in the northern hemisphere countries and have more investment assuring quality and the continuity of data collecting. As a result, in regions like Europe, US and China the high quality long-term data allows for studies using more stations and going far back in time. This is unfortunately not yet the case for South America. But we expect that follow up studies will keep contributing to the understanding of the topic in the region in the future.

---

## Author Response (AR2)

Dear Dr. Stelios Kazadzis,

Thank you very much for the comments on the manuscript "Trends in observed surface solar radiation and their causes in Brazil in the first two decades of the 21st century". We are again thankful for your time and constructive suggestions to improve the quality of the manuscript. We tried to take all your suggestions into consideration. The changes are highlighted in the manuscript.

Here is a point by point response to the editor's comments.

Color code:
Red: Editor's comment.
Black: authors' response.

**Comment from the editor**

The authors have replied in a large number of comments raised by the three referees.

In my opinion is very difficult to accurately assess the role of different factors affecting the SSR.

In the particular study the first issue was to simulate all sky and cloudless sky time series. Temporal aspects play here a very important role on the day by day or month by month calculation of SSR for all and cloudless conditions.

In addition, cloud fraction provides an indication of changes in the solar transmission due to clouds but as the reviewers mention, there are also other cloud properties involved, especially when studying such a large area.

For cloudless cases aerosol play the most important role and it would help to discuss a clear distinction on biomass burning (mostly Amazon related) aerosol changes and anthropogenic/urban changes.

Finally, is not always clear what is the equivalent effect in SSR of a certain per cent change of cloudiness compared with the one based on AOD.

To this aspect, we estimated the Cloud Cover Radiative Effect (CCRE), introduced on section 2.1 and displayed for every composite on table 1. This variable estimates the change in SSR resulting from the observed change in cloud cover.

These three aspects cloudiness – anthropogenic/urban aerosols – biomass burning aerosols, I would try to discuss a bit more clearly in the conclusions.

So in general I think it should be clear that authors:

- Showed what is really happening on SSR in different regions in Brazil based on ground-based measurements

- They used state of the art methods with a number of assumptions that need both objective "limitations" but also subjective ones based on the data and the physics that need to be explained.

- Provided an assessment in some cases with lower and some cases with larger uncertainty, on the causes of SSR changes.

A different approach would be a station-by-station analysis together with radiative transfer modelling to individually assess these results. However, in the current approach a spatial large scale "picture" of SSR and possible effect is provided. And also authors decided to treat each station in exactly the same way concerning the data and methods used for the analysis.

So me recommendation is to try to discuss all the above in the conclusions of the work.

In addition, I would say that there is room for future work especially on quantifying each of the factors causing SSR changes. But in this phase the work can be published with minor revisions mentioned in this document.

We thank the editor for the clear and constructive feedback. We did various minor changes in the conclusion section (all highlighted in the manuscript) to try to address the suggestions. We state more clearly that there is room for further work in the topic. We also try to be more explicit when referring to anthropogenic/urban or biomass burning aerosols. We also mention now explicitly the quality control of the data, which was discussed in more details in section 2.

Concerning technical comments

I would agree that fig. 5 and 7 should mention (not only in the caption) : Lat, lon, and e.g. "AAOD change per decade and %AAOD change per decade" in the axis figures so it is easy for the reader.

We included "AOD per decade" and "AAOD per decade" in the figures, following exactly the suggestion.

Based on comment of reviewer 2 on related literature

I would suggest to try to include more recent papers e.g. Europe

Natsis, A.; Bais, A.; Meleti, C. Trends from 30-Year Observations of Downward Solar Irradiance in Thessaloniki, Greece. Appl. Sci. 2024, 14, 252. https://doi.org/10.3390/app14010252

Kazadzis, S., Founda, D., Psiloglou, B. E., Kambezidis, H., Mihalopoulos, N., Sanchez-Lorenzo, A., Meleti, C., Raptis, P. I., Pierros, F., and Nabat, P.: Long-term series and trends in surface solar radiation in Athens, Greece, Atmos. Chem. Phys., 18, 2395–2411, https://doi.org/10.5194/acp-18-2395-2018, 2018.

These are recent and quite long time measurements.

Also

Schwarz, M.; Folini, D.; Yang, S.; Allan, R.P.; Wild, M. Changes in atmospheric shortwave absorption as important driver of dimming and brightening. Nat. Geosci. 2020, 13, 110–115.

And an older one but still important

Xia, X.; Chen, H.; Li, Z.; Wang, P.; Wang, J. Significant reduction of surface solar irradiance induced by aerosols in a suburban region in northeastern China. J. Geophys. Res. Atmos. 2007, 112, 1–9.

We thank the editor for the paper suggestions, especially for the Schwarz et al. (2020). This is a very important reference in the context of the present study, and it is a known paper, but somehow it was not included in the previous versions of the manuscript. This and all the other suggested papers are now included in the new version of the manuscript.

Also about the comment:

"Lines 70-84: it is necessary to add more information about the steps to move from the hourly values to the annual values in order to prove the robustness of the applied method. If any you can also add a reference of another paper where this method has been applied."

I would agree that this is a very important issue and in addition to the citations to other works ( e.g. Manara et al) a dedicated small paragraph with more details should be added. This would help a lot the reader to assess the presented results.

In the new version of the manuscript we separated one paragraph only describing the procedure from hourly to annual values. We included some additional information from the previous version and now any reader should have enough information to reproduce every step of the procedure we used in the present study.